# Recurrent neural networks enable design of multifunctional synthetic human gut microbiome dynamics

**Mayank Baranwal[1,2†], Ryan L Clark[3†], Jaron Thompson[4], Zeyu Sun[5], Alfred O Hero[5,6,7]\*, Ophelia S Venturelli[3,4,8]\***

[1]Department of Systems and Control Engineering, Indian Institute of Technology, Bombay, India; [2]Division of Data & Decision Sciences, Tata Consultancy Services Research, Mumbai, India; [3]Department of Biochemistry, University of Wisconsin-Madison, Madison, United States; [4]Department of Chemical & Biological Engineering, University of Wisconsin-Madison, Madison, United States; [5]Department of Electrical Engineering & Computer Science, University of Michigan, Ann Arbor, United States; [6]Department of Biomedical Engineering, University of Michigan, Ann Arbor, United States; [7]Department of Statistics, University of Michigan, Ann Arbor, United States; [8]Department of Bacteriology, University of Wisconsin-Madison, Madison, United States

**\*For correspondence:**
hero@eecs.umich.edu (AOH);
venturelli@wisc.edu (OSV)

[†]These authors contributed equally to this work

**Abstract** Predicting the dynamics and functions of microbiomes constructed from the bottom-up is a key challenge in exploiting them to our benefit. Current models based on ecological theory fail to capture complex community behaviors due to higher order interactions, do not scale well with increasing complexity and in considering multiple functions. We develop and apply a long short-term memory (LSTM) framework to advance our understanding of community assembly and health-relevant metabolite production using a synthetic human gut community. A mainstay of recurrent neural networks, the LSTM learns a high dimensional data-driven non-linear dynamical system model. We show that the LSTM model can outperform the widely used generalized Lotka-Volterra model based on ecological theory. We build methods to decipher microbe-microbe and microbe-metabolite interactions from an otherwise black-box model. These methods highlight that Actinobacteria, Firmicutes and Proteobacteria are significant drivers of metabolite production whereas *Bacteroides* shape community dynamics. We use the LSTM model to navigate a large multidimensional functional landscape to design communities with unique health-relevant metabolite profiles and temporal behaviors. In sum, the accuracy of the LSTM model can be exploited for experimental planning and to guide the design of synthetic microbiomes with target dynamic functions.

## Editor's evaluation

The ultimate goal of this work is to apply machine learning to learn from experimental data on temporal dynamics and functions of microbial communities to predict their future behavior and design new communities with desired functions. Using a significant amount of experimental data, the authors suggest a method that outperforms the state-of-the-art approach. The work is of broad interest to those working on microbiome prediction and design.

## Introduction

Microbial communities perform chemical and physical transformations to shape the properties of nearly every environment on Earth from driving biogeochemical cycles to mediating human health and disease. These functions performed by microbial communities are shaped by a multitude of abiotic and biotic interactions and vary as a function of space and time. The complex dynamics of microbial communities are influenced by pairwise and higher order interactions, wherein interactions between pairs of species can be modified by other community members (*Sanchez-Gorostiaga et al., 2019*; *Mickalide and Kuehn, 2019*; *Hsu et al., 2019*). In addition, the interactions between community members can change as a function of time as the community continuously reacts to and modifies its environment (*Hart et al., 2019*). Therefore, flexible modeling frameworks that can capture the complex and temporally changing interactions that determine the dynamic behaviors of microbiomes are needed. These predictive modeling frameworks could be used to guide the design of interventions to precisely manipulate community-level functions to our benefit.

The generalized Lotka-Volterra (gLV) model has been widely used to predict community dynamics and deduce pairwise microbial interactions shaping community assembly (*MacArthur, 1970*). For example, the gLV model has been used to predict the assembly of tens of species based on absolute abundance measurements of lower species richness (i.e. number of species) communities (*Venturelli et al., 2018*; *Mounier et al., 2008*; *Clark et al., 2021*). The parameters of the gLV model can be efficiently inferred based on properly collected absolute abundance measurements and can provide insight into significant microbial interactions shaping community assembly (*Bucci et al., 2016*). However, this model does not represent higher order interactions or microbial community functions beyond species growth. To capture such microbial community functions, composite gLV models have been developed to predict a community-level functional activity based on species abundance at an endpoint (*Clark et al., 2021*; *Stein et al., 2018*). However, these approaches have been limited to the prediction of a single community-level function at a single time point. Therefore, new modeling frameworks are needed to capture temporal changes in multiple community-level functions, such as tailoring the metabolite profile of the human gut microbiome (*Fischbach and Sonnenburg, 2011*).

Neural network architectures, such as recurrent neural networks (RNNs), are universal function approximators (*Dambre et al., 2012*; *Schäfer and Zimmermann, 2006*) that enable greater flexibility compared to gLV models for modeling dynamical systems. However, neural network based models often require significantly more model parameters, which poses additional challenges to model fitting and generalizability. A particular RNN model architecture called long short-term memory (LSTM) addresses challenges associated with training on sequential data by incorporating gating mechanisms that learn to regulate the influence of information from previous instances in the sequence (*Lipton et al., 2015*). From their initial successes in speech recognition (*Graves et al., 2005*) and computer vision (*Byeon et al., 2015*), LSTMs have recently been applied to modeling biological data such as subcellular localization of proteins (*Sonderby et al., 2015*) and prediction of biological age from activity collected from wearable devices (*Rahman and Adjeroh, 2019*). Related to microbiomes, deep learning frameworks have been applied to predict gut microbiome metabolites based on community composition data (*Le et al., 2020*), final community composition based on microbial interactions (*Larsen et al., 2012*) and end-point community composition based on the presence/absence of species (*Michel-Mata et al., 2021*). In addition, RNN architectures have been used to model phytoplankton (*Jeong et al., 2001*) and macroinvertebrate (*Chon et al., 2001*) community dynamics. Despite achieving reasonable prediction performance, previous efforts at modeling ecological system dynamics using RNNs are typically limited to handful of organisms (<10), have provided limited model interpretation and have not been leveraged to predict temporal changes in community behaviors. In addition, RNN architectures have not been used for bottom-up community design, which could be exploited for applications in bioremediation, bioprocessing, agriculture, and human health (*Leggieri et al., 2021*; *Lawson et al., 2019*; *Clark et al., 2021*).

Here, we apply LSTMs to model time dependent changes in species abundance and the production of key health-relevant metabolites by a diverse 25-member synthetic human gut community. LSTMs are a good model for microbiomes because (1) LSTMs are a natural choice for a neural network based model of time-series data (*Goodfellow et al., 2016*); (2) LSTMs are highly flexible models that can capture complex interaction networks that are often neglected in ecological models; (3) LSTMs can be modified to capture additional system variables such as environmental factors (e.g. metabolites).

In addition, LSTMs have some advantages over traditional RNNs because they can capture long-term dependencies. LSTMs have additional parameters that adjust the effects of earlier time points on the predictions at later time points in a time-series. We use the trained LSTM model to elucidate significant microbe-microbe and microbe-metabolite interactions.

The flexibility and accuracy of the LSTM model enabled systematic integration into our experimental planning process in two stages. First, the LSTM was fit to data from a previous study with low temporal resolution involving a moderate number of synthetic microbial communities (*Clark et al., 2021*). The distribution of LSTM metabolite predictions was then used to identify sparse sub-communities in the tails of the distribution, communities that we refer to as 'corner cases'. A second experiment was then performed that expanded the training data for the LSTM in the vicinity of these corner cases with higher time resolution. The LSTM-guided two-stage experimental planning procedure substantially reduced the number of experiments compared to random sampling of the functional landscape with temporal resolution in a single stage experiment. Therefore, the LSTM analysis enabled our main findings on dynamical behaviors of communities and identified the key species critical for community assembly and metabolite profiles. Compared to the gLV model, the proposed LSTM framework provides a better fit to the experimental data, captures higher order interactions and provides higher accuracy predictions of species abundance and metabolite concentrations. In addition, our approach preserves model interpretability through a suitably developed gradient-based framework and locally interpretable model-agnostic explanations (LIME) (*Ribeiro et al., 2016a*). Using our time-series data of species abundance and metabolite concentrations, we demonstrate that the temporal behaviors of the communities cluster into distinct groups based on the presence and absence of sets of species. Our results highlight that LSTM models are powerful tools for predicting and designing the dynamic behaviors of microbial communities.

## Results

### LSTM outperforms the generalized Lotka Volterra ecological model

Our first objective was to compare the predictive performance of the LSTM model to a commonly used ecological modeling approach. The gLV model is a widely used ecological model consisting of a coupled set of ordinary differential equations that captures the growth dynamics of members of a community based on their intrinsic growth rate and interactions between all pairs of community members (*Venturelli et al., 2018*). Therefore, the gLV model is not suited to capture higher order interactions among species or changes in inter-species interactions resulting from variation in the environment. By contrast, the LSTM modeling framework is flexible and can capture complex relationships between species as well as time-dependent changes in inter-species interactions. To evaluate the strengths and limitations of these modeling frameworks, we characterized the performance of the gLV and LSTM models in learning the behavior of a ground truth model that included pairwise and third-order interactions between species (Methods).

Our ground truth model is based on a gLV model of a 25-member synthetic gut community from a previous study (*Clark et al., 2021*). To perturb our ground truth model with higher order interactions, we add third-order interaction terms with either mild or moderate parameter magnitudes (Methods). Using this model, we simulate sub-communities that vary in the number of species. Of all the randomly simulated communities, those containing six or fewer species are used to train both the gLV and LSTM models (624 training communities), while the remaining communities (3299 test communities with ≥10 species) are used as a hold-out test set. The 624 training communities includes 25 monospecies, 300 unique pairwise communities, 100 unique three-member communities, 100 unique five-member communities, and 99 unique six-member communities. The simulated data spans 48 hr separated by an interval of 8 hr, reflecting the experimentally feasible periodic sampling interval of 8 hr.

Recall that we restrict our attention to simpler (fewer species) communities for training to determine if the behavior of lower order communities can be used to predict higher order communities. Further, pairwise inter-species interactions are easier to decipher in lower order communities due to potential co-variation among parameters (correlations between parameters) as a consequence of model structure or methods of data collection. A similar training/test partitioning was used to generate predictive models of complex community behaviors (*Venturelli et al., 2018*; *Clark et al., 2021*; *Hromada et al., 2021*).

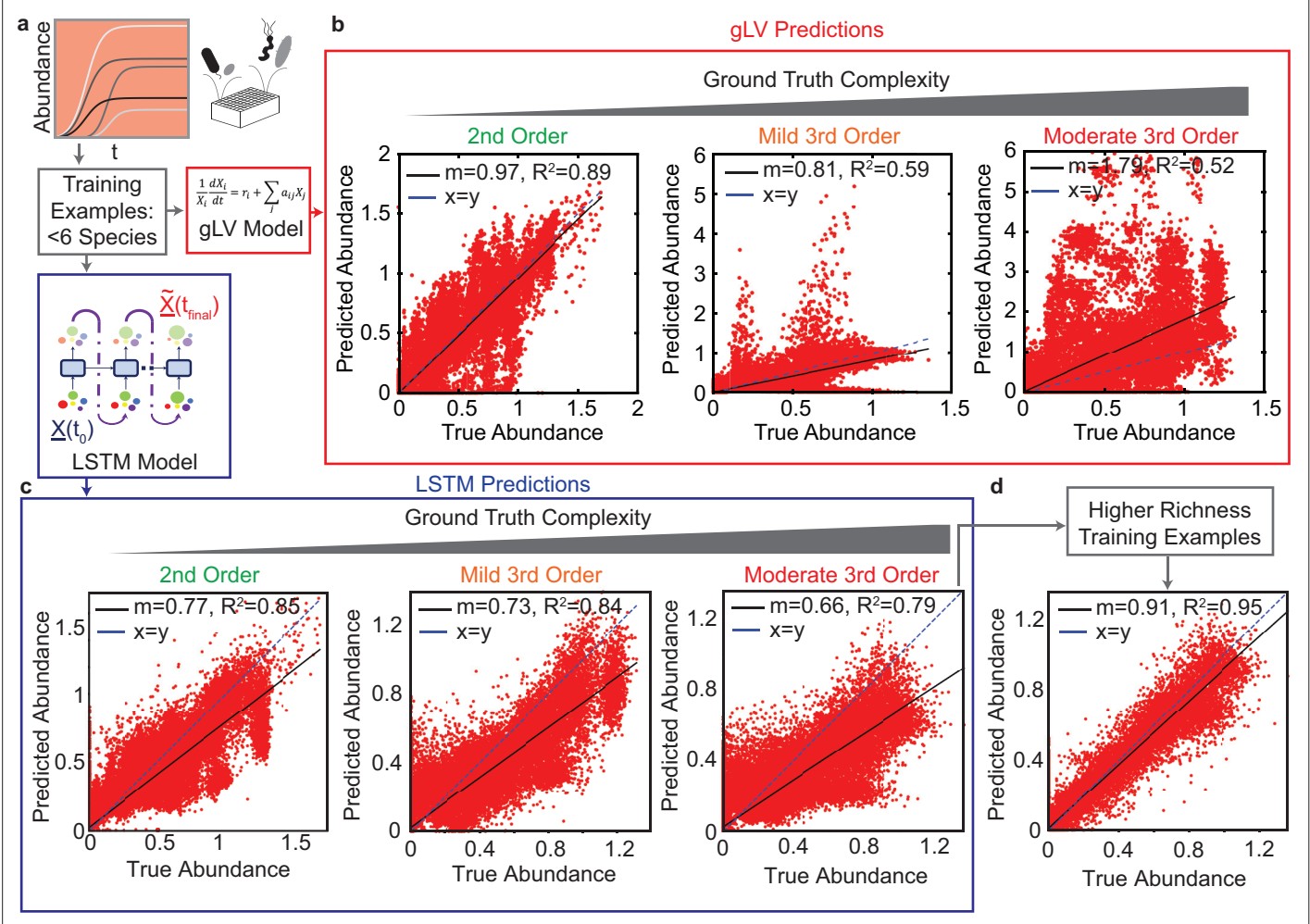

**Figure 1.** Comparison of generalized Lotka Volterra (gLV) and Long Short Term Memory (LSTM) model prediction performance of species abundance in a 25-member microbial community in response to third-order perturbations of varying magnitude. For both models, training data consists of low species richness communities ($> 0.05$ species, $N = 82,475$, Pearson correlation p-value $lt_{0.0001}$). (**a**) & (**d**): Data was generated using a gLV model that captures monospecies growth and pairwise interactions. Scatter plots of true versus predicted species abundance at $t = 48 \text{hr}$ using the gLV and LSTM models, respectively. $X$ represents a vector of species abundances. (**b**) & (**e**) Scatter plot of true versus predicted species abundance of the gLV and LSTM models, respectively when the simulated data is subjected to low magnitude (mild) third-order interactions. (**c**) & (**f**) Scatter plot of true versus predicted species abundance of gLV and LSTM models, respectively when the simulated data is further subjected to moderately large third-order interactions. (**g**) Scatter plot of true versus predicted species abundance for the LSTM model. The training set included a set of higher richness communities (50 each of 11 and 19 member communities). All predictions are forecasted from the species abundance at time 0.

The prediction performance of the trained gLV and LSTM models on the hold-out test set are similar for the ground truth model containing only pairwise interactions (Pearson $R^2$ of 0.89 and 0.85 for gLV and LSTM models, respectively) (**Figure 1b, c** left). For the ground truth model with mild third-order interactions (interaction coefficients that do not exceed 25% of the maximum of the absolute values of the coefficients for the second-order interactions), the performance of the LSTM model is substantially better than the gLV model with the $R^2$-score of 0.85, as opposed to 0.52 for the gLV model (**Figure 1b, c**, middle). In addition, the LSTM model performs significantly better than the gLV model for higher magnitude (moderate) third-order perturbations (third-order interaction coefficients that do not exceed 50% of the maximum of the absolute values of the coefficients for second-order interactions) (**Figure 1b, c**, right).

This in silico analysis highlights the advantages of adopting more expressive neural network models over severely constrained ecological models such as gLV. In addition, a key advantage of the proposed LSTM model over the gLV model is the amount of time required for training the two models. The gLV equations are coupled nonlinear ordinary differential equations, and thus training gLV models requires

substantial computational time (nearly 5–6 hr), whereas the LSTM models can be trained in minutes on the same platform. Therefore, the LSTM approach is highly suited for real-time training and planning of experiments. Note that both the composite as well as the LSTM model require tuning of hyperparameters for optimal performance. The details of the computational implementation are provided in the Methods section.

To further leverage this in silico experimental approach, we aimed to identify what type of datasets are required for building predictive models of high richness community behaviors depending on the nature of their underlying interactions. In further analyzing our results, we observed a crescent shaped prediction profile, representing an inherent bias, which we hypothesized was due to the training data containing only communities with ≤6 species (*Figure 1c*). To test this hypothesis, we augmented the training set with 100 communities enriched with a larger number of species (randomly sampled 11 and 19-member communities). Using this enriched training set, the LSTM model accurately predicts the community dynamics of the hold-out set with an $R^2$ of 0.95 (*Figure 1d*). In sum, the LSTM has difficulty predicting the behavior of high richness communities when the training data consists of only low richness communities. However, adding a moderate number of high richness communities to the training set eliminates the prediction bias and improves the prediction performance of the LSTM.

## LSTM accurately predicts experimentally measured microbial community assembly

After validating our methods using the ground truth modeling approach described above, we evaluated the ability of the LSTM to capture the dynamics of experimentally characterized synthetic human gut microbial communities. We tested the effectiveness of the LSTM on time-resolved species abundance data from a previous study of a well-characterized twelve-member synthetic human gut community (*Venturelli et al., 2018*). The experimental data consists of species abundance sampled approximately every 12hr. A total of 175 microbial communities with sizes varying from 2 to 12 were used to train and evaluate the LSTM model. Of the 175 microbial communities, 102 microbial communities were selected randomly to constitute the training set, while the remaining 73 microbial communities constituted the hold-out test set (*Supplementary file 1*). This train/test split was similar to that used to train a gLV model in the previous study (*Venturelli et al., 2018*). The previous study represented perturbations in cell densities and nutrient availability by diluting the community 20-fold every 24 hr into fresh media (i.e. passaging of the communities) (*Figure 2a*). The sequential dilutions of the communities are external perturbations that introduce further complexity towards model training.

We trained a LSTM network to predict species abundances at various time points given the information of initial species abundance. We found that a total of five LSTM units can predict species abundance at different time points (12, 24, 36, 48, and 60 hr) based on the initial species abundance. The output of each LSTM unit is used as an input to the next unit. However, the input to the current LSTM unit is randomized between the output from the previous LSTM unit and the true abundance at the current time point in the randomized teacher forcing mode of training in order to eliminate temporal bias in the prediction of end-point species abundances. We did not model the passaging perturbations explicitly, since the experimental procedure was consistent across all communities. This also highlights the advantage of using black-box approaches, such as the LSTM network, where physical parameters such as dilution do not need to be explicitly modeled. Here, each LSTM unit consists of a single hidden layer comprising of 2048 hidden units with ReLU activation. The details on hyperparameter tuning, learning rates, and choice of optimizer are provided in the Methods section.

Despite the passaging perturbations and variation in the sampling times, the LSTM accurately predicts (Pearson $R^2$-scores of 0.74, 0.73, 0.74, 0.70, and 0.69 at time points 12, 24, 36, 48, and 60 hr, respectively) not only the end-point species abundance, but also the abundances at intermediate time points on hold-out test sets (*Figure 2b-f*). These results demonstrate that the LSTM model can accurately predict the temporal changes in species abundance of multi-species communities in the presence of external perturbations. Representative communities that were accurately or poorly predicted by the LSTM are shown in (*Figure 2—figure supplement 1*).

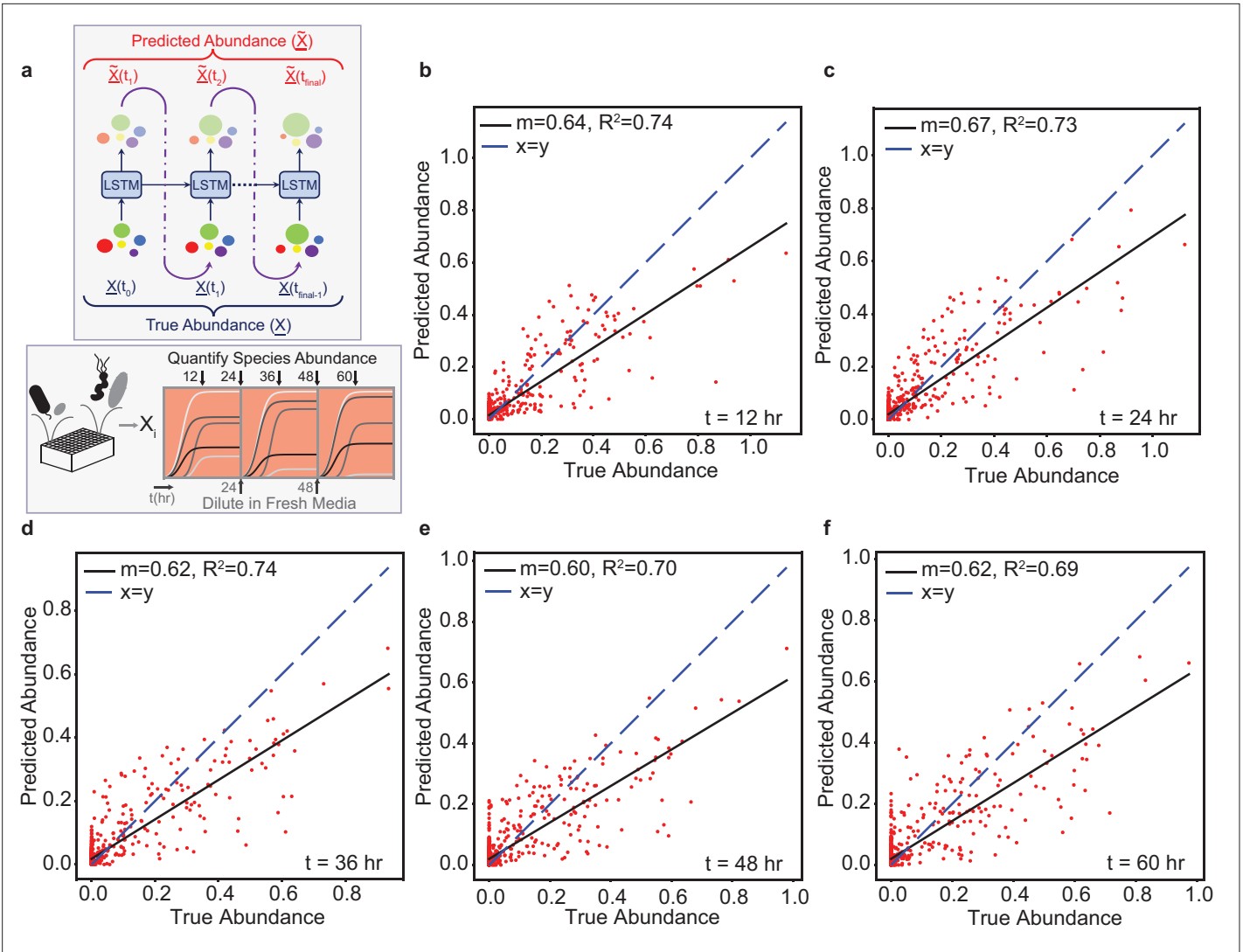

**Figure 2.** The LSTM model can predict the temporal changes in species abundance in a 12-member synthetic human gut community in response to periodic dilution (passaging). (**a**) Proposed LSTM modeling methodology for the dynamic prediction of species abundance in a microbial community. The initial abundance information is an input to the first LSTM cell, the output of which is trained to predict abundance at the next time point. Consequently, the predicted abundance becomes an input to another LSTM cell with shared weights to predict the abundance at the subsequent time point. The process is repeated until measurements at all time points are available. $X$ represents a vector of species abundances. Thus, all predictions are forecasted from the abundance at time 0. (**b**) Scatter plot of measured (true) and predicted species abundance of a 12-member synthetic human gut community at 12 hr ($N = 876$, p-value $= 2.44e - 257$). (**c**) Scatter plot of measured (true) and predicted abundance at 24 hr (p-value $= 6.51e - 257$). (**d**) Scatter plot of measured (true) and predicted abundance at 36 hr (p-value $= 7.42e - 257$). (**e**) Scatter plot of measured (true) and predicted abundance at 48 hr (p-value $= 1.66e - 227$). (**f**) Scatter plot of measured (true) and predicted abundance at 60 hr (p-value $= 3.39e - 227$).

The online version of this article includes the following figure supplement(s) for figure 2:

**Figure supplement 1.** Prediction of temporal changes of species abundance for a few representative communities by the LSTM network.

## LSTM enables end-point design of multifunctional synthetic human gut microbiomes

The chemical transformations (i.e. functions) performed by the community are the key design variables for microbiome engineering goals, as evidenced by their major impacts on human health (*Sharon et al., 2014*). Thus, we explored prediction of microbial community functions by applying the LSTM framework to design health-relevant metabolite profiles using synthetic human gut communities.

A core function of gut microbiota is to transform complex dietary substrates into fermentation end products such as the beneficial metabolite butyrate, which is a major determinant of gut homeostasis

(*Litvak et al., 2018*). In a previous study, we designed butyrate-producing synthetic human gut micro-biomes from a set of 25 prevalent and diverse human gut bacteria using a composite gLV and statistical model (*Clark et al., 2021*). While the composite model approach was successful in predicting butyrate concentration, designing community-level profiles of multiple metabolites adds substantial complexity and limited flexibility using the composite modeling approach. Thus, we leveraged the accuracy and flexibility of LSTM models to design the metabolite profiles of synthetic human gut microbiomes. We focused on the fermentation products butyrate, acetate, succinate, and lactate which play important roles in the gut microbiome's impact on host physiology and interactions with constituent community members (*Fischbach and Sonnenburg, 2011*).

We used the species abundance and metabolite concentrations from our previous work (*Clark et al., 2021*) to train an initial LSTM model. This model uses a feed-forward network (FFN) at the output of the final LSTM unit that maps the endpoint species abundance (a 25-dimensional vector) to the concentrations of the four metabolites (*Figure 3a*). The entire neural network model comprising LSTM units and a feed-forward network is learned in an end-to-end manner during the training process, (i.e. all the network weights are trained simultaneously). Cross-validation of this model (Model M1, *Supplementary file 1*) on a set of hold-out community observations shows good agreement between the model predictions and experimental measurements for metabolite concentrations and microbial species abundances (*Figure 3—figure supplement 1*). Thus, we used this model to design high species richness (i.e. >10 species) communities with tailored metabolite profiles (*Figure 3a*).

We first used the LSTM model M1 to simulate every possible combination of >10 species (26,434,916 total communities). The simulated communities separate into two regions: one centered around a dense ellipse of high butyrate concentration characterized by communities containing the butyrate-producing species *Anaerostipes caccae* (AC) and a second dense ellipse of communities that produce low levels of butyrate and lacked AC (*Figure 3b*). This bimodality due to the presence/absence of AC is consistent with our previous finding that AC is the strongest driver of butyrate production in this system (*Clark et al., 2021*). In addition, the strong negative correlation between lactate and butyrate in the AC+ cluster of communities ($R^2 = 0.72$, $p < 0.001$, N=14,198,086) is consistent with the ability of AC to transform lactate into butyrate (*Clark et al., 2021*). These results demonstrate that the LSTM model can capture the major microbial drivers of metabolite production as well as the correlations between different metabolites.

We used our simulated metabolite production landscape to plan informative experiments for testing the predictive capabilities of our model. First, we designed a set of 'distributed' communities that spanned the range of typical metabolite concentrations predicted by our model. To this end, we selected 100 communities closest to the centroids of 100 clusters determined using k-means clustering of the four-dimensional metabolite space. Second, we designed a set of communities to test our model's ability to predict extreme shifts in metabolite outputs. To do so, we identified four 'corners' of the distribution in the lactate and butyrate space (*Figure 3b*). We next examined the relationship between acetate and succinate within each of these corners and found that the distributions varied depending on the given corner (*Figure 3b*, inset). The total carbon concentration in the fermentation end products across all predicted communities displayed a narrow distribution (mean 316 mM, standard deviation 20 mM, *Figure 3—figure supplement 2*). The production of the four metabolites are coupled due to the structure of metabolic networks and fundamental stoichiometric constraints (*Oliphant and Allen-Vercoe, 2019*). Therefore, the model learned the inherent 'trade-off' relationships between these fermentation products based on the patterns in our data. We chose a final set of 80 'corner' communities for experimental validation (five communities from each combination of maximizing or minimizing each metabolite, Methods).

By experimentally characterizing the endpoint community composition and metabolite concentrations of the 180 designed communities, we found that the LSTM model M1 accurately predicted the rank order of metabolite concentrations and microbial species abundances. The LSTM model substantially outperformed the composite model (gLV and regression, model from previous work [*Clark et al., 2021*]) trained on the same data for the majority (59%) of output variables (*Figure 3—figure supplement 3a*). Additionally, replacing the regression module of the composite model with either a Random Forest Regressor or a Feed Forward Network did not improve the metabolite prediction accuracy beyond that of the LSTM (*Figure 3—figure supplement 3a*). One of the key limitations of the composite models is that the metabolite variables are a function of the endpoint species

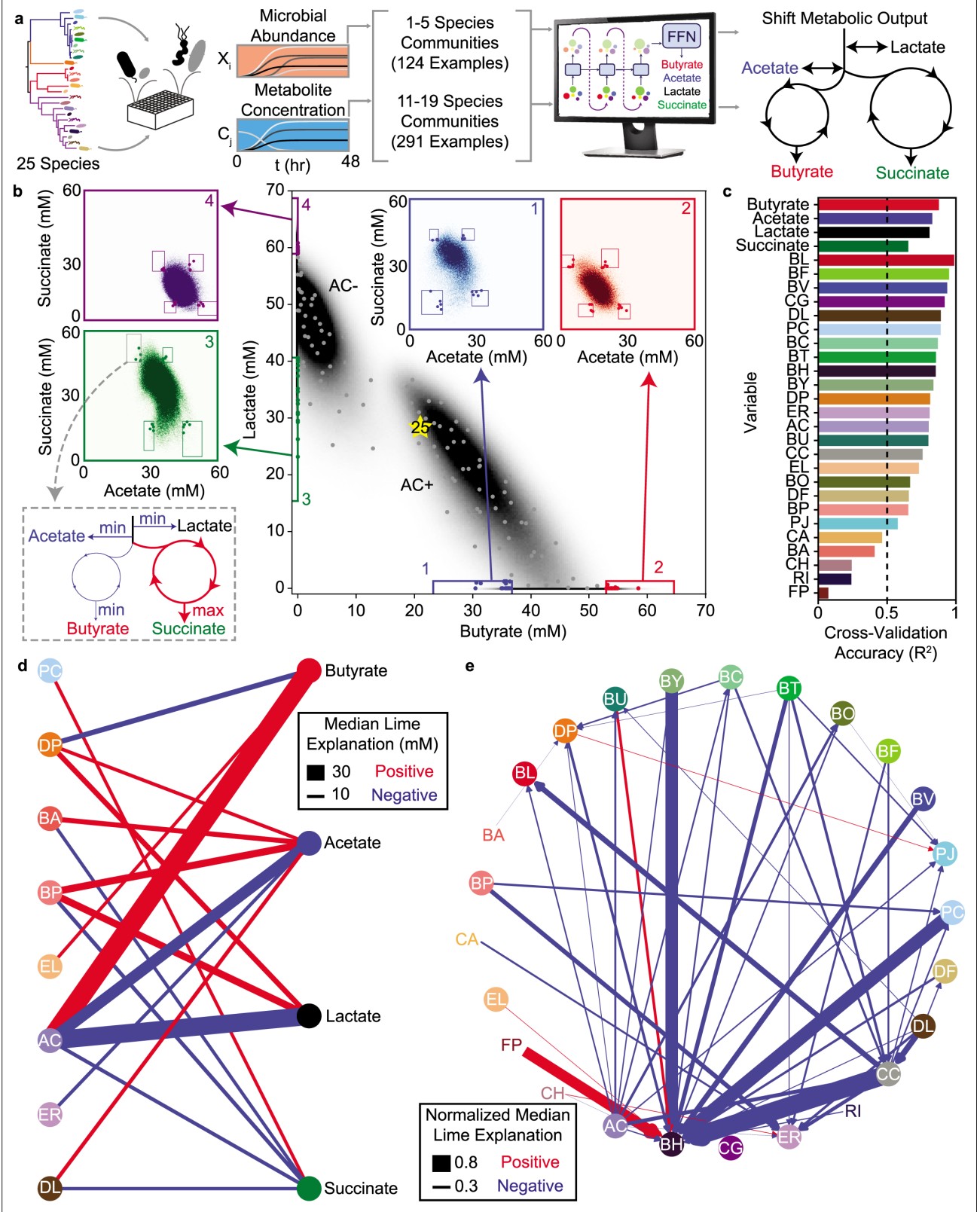

**Figure 3.** LSTM-guided design and interpretability of community-level metabolite production profiles (**a**) Schematic of model-training and design of communities with tailored metabolite outputs. (**b**) Heatmap of butyrate and lactate concentrations of all possible communities predicted by the LSTM model M1. Grey points indicate communities chosen via $k$-means clustering to span metabolite design space. Colored boxes indicate 'corner' regions defined by $95^{th}$ percentile values on each axis with points of the corresponding color indicating designed communities within that 'corner'.

*Figure 3 continued*

Insets show heat maps of acetate and succinate concentrations for all communities within the corresponding boxes on the main figure. Boxes on the inset indicate 'corners' defined by $95^{th}$ percentile values on each axis with colored points corresponding to the same points indicated on the main plot. (**c**) Cross-validation accuracy of LSTM model trained and validated on a random 90/10 split of all community observations (model M2), evaluated as Pearson correlation $R^2$ for the correlation of predicted versus measured for each variable (all p-values$It_{0.05}$, N and p-value for each test reported in ***Supplementary file 1***). Dashed line indicates $R^2 = 0.5$, which is used as a cutoff for including a variable in the subsequent network diagrams. (**d**) and (**e**) Network representation of median LIME explanations of the LSTM model M2 from (**c**) for prediction of each metabolite concentration (**d**) or species abundance (**e**) by the presence of each species. Edge widths are proportional to the median LIME explanation across all communities from (**b**) used to train the model in units of concentration (for (**d**)) or normalized to the species' self-impact (for (**e**)). Only explanations for those variables where the cross-validated predictions had $R^2 > 0.5$ are shown. Networks were simplified by using lower thresholds for edge width (5 mM for (**d**), 0.2 for (**e**)). Red and blue edges indicate positive and negative contributions, respectively.

The online version of this article includes the following figure supplement(s) for figure 3:

**Figure supplement 1.** Cross-validation of LSTM model M1 predictions of species abundance and metabolite concentration.

**Figure supplement 2.** Predicted total carbon in fermentation products.

**Figure supplement 3.** Prediction and classification statistics for model M1 predictions of designed community sets.

**Figure supplement 4.** Metabolite production of each species grown in monoculture Bars show the mean net production or consumption of each metabolite for monocultures of each species (bar color indicates species as specified in the legend).

**Figure supplement 5.** Metabolite-species LIME explanations computed over a 20-fold partitioning of the data set.

**Figure supplement 6.** Microbe-microbe LIME explanations computed over a 20-fold partitioning of the data set.

**Figure supplement 7.** Comparison of LIME explanations of LSTM to gLV Parameters.

abundance, but the species abundances are not a function of the metabolite concentrations. By contrast, the LSTM model can capture such feedbacks between metabolites and species. Notably, the LSTM model prediction accuracy for the metabolites was similar for both the 'distributed' and 'corner' communities (***Figure 3—figure supplement 3b–e***). These results indicate that our model is useful for designing communities with a broad range of metabolite profiles that includes those at the extremes of the metabolite distributions.

To determine if the LSTM model could separate groups of communities with extreme behaviors, we treated the 'corners' as classes and quantified the classification accuracy of our model. The model accurately classified the communities when considering only butyrate and lactate concentrations. However, the model had poorer separation when acetate and succinate were also considered in defining the classes (***Figure 3—figure supplement 3f***). The misclassification rate was higher for small Euclidean distances between classes and decreased with the Euclidean distance (***Figure 3—figure supplement 3g***). This implies that the insufficient variation in concentrations due to fundamental stoichiometric constraints limited our ability to define 16 distinct classes that maximized/minimized each metabolite. While model M1 accurately predicted metabolite concentrations and the majority of species abundances, several individual species abundances were poorly predicted ($R^2 = 0 - 0.6$, ***Figure 3—figure supplement 3a***). Thus, we used the dataset to improve the LSTM model. To this end, we combined the new observations with the original observations and randomly partitioned the data into 90% for training and 10% for cross-validation. The resulting model (M2, ***Supplementary file 1***) was substantially more predictive of species abundances ($R^2 > 0.5$ for all but five species FP, RI, CA, BA, CH (***Figure 3c***)).

## Using local interpretable model-agnostic explanations to decipher interactions

One of the commonly noted limitations of machine learning models is their lack of interpretability for extracting biological information about a system. Fortunately, generally applicable tools have been developed to aid in model interpretation. Thus, we sought to use such methods to decipher key relationships among variables within the LSTM to deepen our biological understanding of the system. We used local interpretable model-agnostic explanations (LIME) (***Ribeiro et al., 2016b***), to quantify the impact of each species' presence on each metabolite and species in each of the sub-communities used to train model M2. We used the median impact of each species presence on each metabolite or species across all training instances to generate networks that revealed microbe-metabolite (***Figure 3d***) and microbe-microbe (***Figure 3e***) interactions. In general, these networks represent broad

design principles for the community metabolic outputs by indicating which species have the most consistent and strong impacts on each metabolite and species abundance across a wide range of sub-communities. For instance, the metabolite network highlights *Anaerostipes caccae* (AC) as having the largest positive effect on butyrate production with an additional positive contribution from EL and a negative contribution from DP, consistent with the previous composite gLV model of butyrate production by this community (*Clark et al., 2021*).

In addition, the number of microbial species impacting each metabolite in these networks trended with the number of microbial species that individually produced or consumed each metabolite (*Figure 3—figure supplement 4*). For example, butyrate displayed the fewest edges (3) and was produced by the lowest number of individual species (4). By contrast, acetate had the most edges (6) and was produced by the largest number of individual species (19). The inferred microbe-metabolite network consisted of diverse species including Proteobacteria (DP), Actinobacteria (BA, BP, EL), Firmicutes (AC, ER, DL) and one member of Bacteroidetes (PC), but excluded members of *Bacteroides*. Therefore, while *Bacteroides* exhibited high abundance in many of the communities, they did not substantially impact the measured metabolite profiles but instead modulated species growth and thus community assembly (*Figure 3e*). We explored the consistency of LIME explanations for the full 25-member community in response to random partitions of the training data to provide insights into the sensitivity of the LIME explanations given the training data (*Figure 3—figure supplement 5*, *Figure 3—figure supplement 6*). These results demonstrated that the direction of the strongest LIME explanations of the full community were consistent in sign despite variations in magnitude. One exception is for the species *Roseburia intestinalis* (RI), which had high variability across different test/train splits. This is consistent with previous observations that RI has substantial growth variability across experimental communities (*Clark et al., 2021*). In sum, these results demonstrate that in general the LIME explanations were robust to variations in the training data.

The LIME explanations of inter-species interactions exhibited a statistically significant correlation with their corresponding inter-species interaction parameters from a previously parameterized gLV model of this system (*Clark et al., 2021*; *Figure 3—figure supplement 7a*). The sign of the interaction was consistent in 80% of the interactions with substantial magnitude (>0.05 in both the LIME explanations and gLV parameters) (*Figure 3—figure supplement 7b*). This consistency with previous observations suggests that the LSTM model was able to capture similar broad trends in inter-species relationships as gLV (interpreted through the average LIME explanation across all observed communities). The LSTM model captured more nuanced context-specific behaviors (interpreted as the LIME explanation for one specific community context) than the mathematically restricted gLV model, which substantially improved the predictive capability of the LSTM model. These results demonstrate that the LSTM framework is useful for developing high accuracy predictive models for the design of precise community-level metabolite profiles. Our approach also preserves the ability to decipher different types of interactions in the LSTM model that are explicitly encoded in less accurate and flexible ecological models such as gLV.

## Sensitivity of LSTM model prediction accuracy highlights poorly understood species and pairwise interactions

Identification of species that limit prediction performance could guide selection of informative experiments to deepen our understanding of the behaviors of poorly predicted communities. Therefore, we evaluated the sensitivity of the LSTM model (model M2) prediction accuracy to species presence/absence and the amount of training data. High sensitivity of model prediction performance to the number of training communities indicates that collection of additional experimental data would continue to improve the model. Additionally, identifying poorly understood communities will guide machine learning-informed planning of experiments. To evaluate the model's sensitivity to the size of the training dataset, we computed the hold-out prediction performance ($R^2$) as a function of the size of the training set by sub-sampling the data (*Figure 4a*). We used 20-fold cross-validation to predict metabolite concentrations and species abundance. Our results show that the ability to improve prediction accuracy as a function of the size of the training data set was limited by the variance in individual species abundance in the training set (*Figure 4—figure supplement 1*). For instance, certain species with low variance (e.g. FP, EL, DP, RI) in abundance in the training set displayed low sensitivity to the amount of training data and were poorly predicted by the model. The high sensitivity of specific

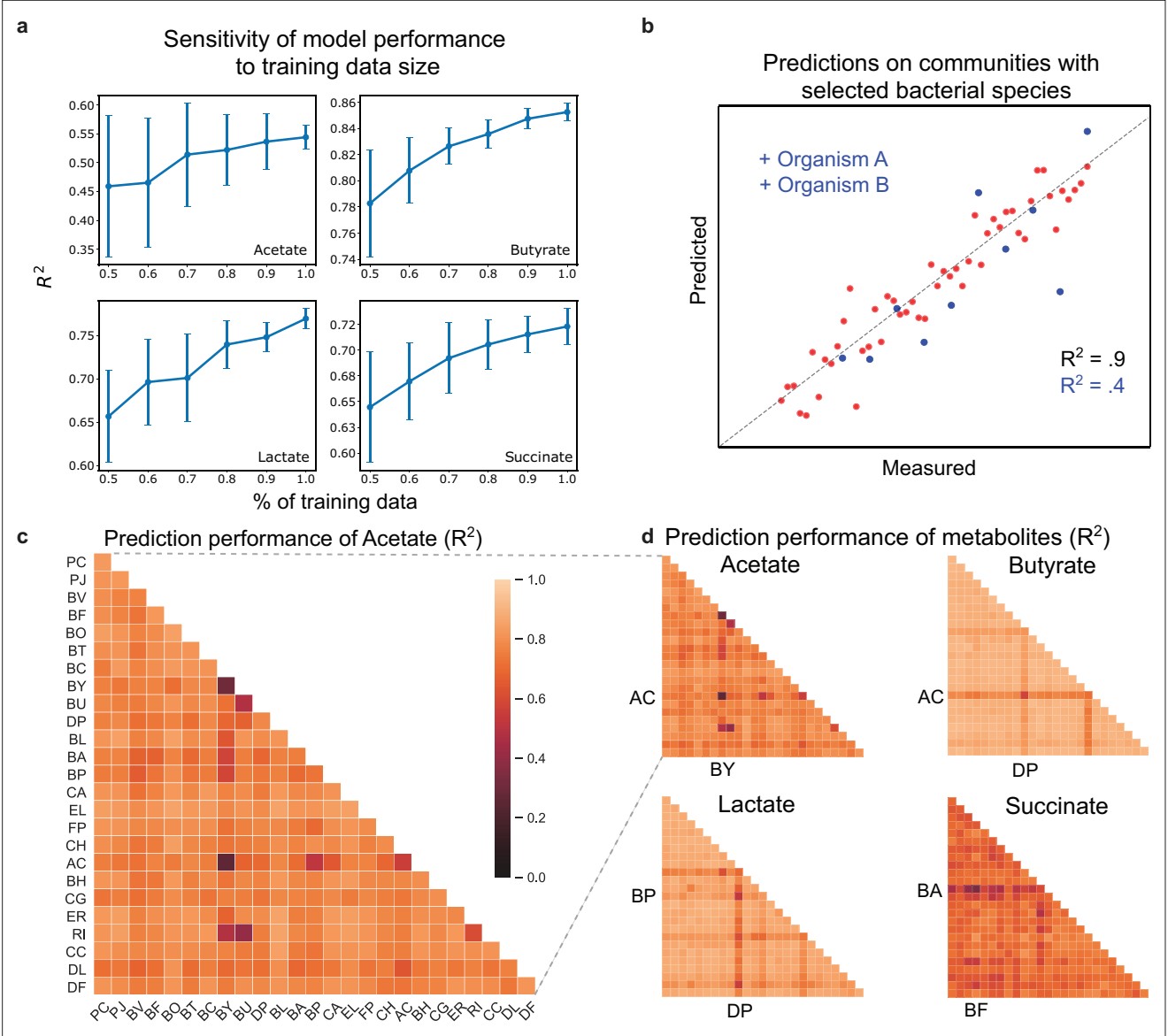

**Figure 4.** Hold-out prediction performance on sub-communities provides information about poorly understood species and interactions between species. (**a**) Sensitivity of metabolite prediction performance ($R^2$) to the amount of training data. Training datasets were randomly subsampled 30 times using 50–100% of the total dataset in increments of 10%. Each subsampled training set was subject to 20-fold cross-validation to assess prediction performance. Lineplot of the mean prediction performance over the 30 trials for each percentage of the data. Error bars denote 1 s.d. from the mean. (**b**) Schematic scatter plot representing how communities containing species A and B define a poorly predicted subsample of the full sample set (**c**) Heatmap of prediction performance ($R^2$) of acetate for each subset of communities containing a given species (diagonal elements) or pair of species (off-diagonal elements). (**d**) Heatmap of prediction performance for acetate, butyrate, lactate, and succinate. A sample subset containing a given species or pair of species included all communities in which the species were initially present. Predictions for each community were determined using 20-fold cross validation so that for each model the predicted samples were excluded from the training samples. N and p-values are reported in *Supplementary file 1*.

The online version of this article includes the following figure supplement(s) for figure 4:

**Figure supplement 1.** Sensitivity of species abundance prediction performance ($R^2$) to the size of the training dataset.

metabolites (e.g. lactate) and species (e.g. AC, BH) to the amount of training data indicates that further data collection would likely improve the model's prediction performance.

To determine how pairwise combinations of species impacted model prediction performance, we used 20-fold cross-validation to evaluate the prediction performance ($R^2$) on subsets of the total dataset, where subsets were selected based on the presence of individual species or pairs of species

(*Figure 4b*). Using this approach, we identified individual species and species pairs that had the greatest impact on the prediction performance of metabolite concentrations. Sample subsets with poor prediction performance highlight individual species and species pairs whose presence reduced the model's ability to accurately predict metabolite concentrations. Although the subsets were smaller than the total data set ($n = 761$), calculation of prediction performance was not limited by small sample sizes, where the number of communities in each subset ranged from $n = 77$ to $n = 478$.

The interaction network shown in *Figure 3d* shows the impact of individual species on each metabolite, but does not provide information about whether the effect is due to individual species or pairwise interactions. To determine whether pairwise interactions influence metabolite concentrations, we quantified how prediction performance changed in response to the presence individual species and pairs of species. Specifically, if prediction performance taken over a subset of communities containing a given species pair was markedly different than prediction performance for the subsets corresponding to the individual species, this implies that the given pairwise interaction impacts metabolite production. Using *equation 5* (Methods), we found that the prediction performance of lactate and butyrate were the least sensitive to species pairs (average decrease in prediction performance for subsets with species pairs of 0.72% and 1.10% compared to corresponding single species subsets). However, the prediction performance of acetate and succinate were the most sensitive to the presence of species pairs (increase in prediction performance of 6.68% for acetate and a decrease of 2.951% for succinate). This difference in prediction performance suggests that pairwise interactions influences the production of acetate and succinate, while the production of lactate and butyrate are primarily driven by the action of single species. The sensitivity of acetate and succinate to pairwise interactions is consistent with the inferred interaction network shown in *Figure 3d*, which highlights multiple species-metabolite interactions for acetate and succinate and sparse and strong species-metabolite interactions for butyrate and lactate.

Pairs of certain *Bacteroides* and butyrate producers including BY-RI, BU-RI, and BY-AC resulted in reduced prediction performance of acetate. This suggests that interactions between specific *Bacteroides* and butyrate producers were important for acetate transformations, which is consistent with the conversion of acetate into butyrate. Based on the LIME analysis in *Figure 3d*, AC, DP, and BP had the largest impact on lactate. Thus, the hold-out prediction performance for lactate was primarily impacted by specific pairs that include these species. In sum, these results demonstrate how the LSTM model can be used to identify informative experiments for investigating poorly understood species and interactions between species, where collection of more data would likely improve model prediction performance.

## Time-resolved measurements of communities reveal design rules for qualitatively distinct metabolite dynamics

We next leveraged the LSTM model's dynamic capabilities to understand the temporal changes in metabolite concentrations and community assembly of the 25-member synthetic gut microbiome. To this end, we chose a representative subset of 95 out of the 180 communities from *Figure 3b*, *Figure 5—figure supplement 1a*, 60 communities for training, 34 for validation, plus the full 25-member community and experimentally characterized species abundance and metabolite concentrations every 16 hr during community assembly (*Figure 5a*). We analyzed the dynamic behaviors of these communities using a clustering technique to extract high-level design rules of species presence/absence that determined qualitatively distinct temporal metabolite trajectories (i.e. broad trends consistent across a set of communities) and exploited the LSTM framework to identify context-specific impacts of individual species on metabolite production (i.e. a more fine-tuned case-by-case analysis).

The temporal trajectories of species abundance and metabolite concentrations showed a wide range of qualitatively distinct trends across the 95 communities (*Figure 5b–g*). For example, some metabolites concentrations monotonically increased (e.g. butyrate in *Figure 5b, c, e and g*), monotonically decreased (e.g. lactate in *Figure 5b, c*) or exhibited biphasic dynamics (e.g. acetate in *Figure 5c*). To determine if there were communities with similar temporal changes in metabolite concentrations, we clustered communities using a minimal spanning tree (*Grygorash et al., 2006*) on the Euclidean distance between the metabolite trajectories of each pair of communities (*Figure 5a*). The resulting six clusters exhibited high quantitative within-cluster similarity and qualitatively distinct metabolite trajectories (*Figure 5b–g*). Clusters 4 and 5, which contained the largest number of communities, had

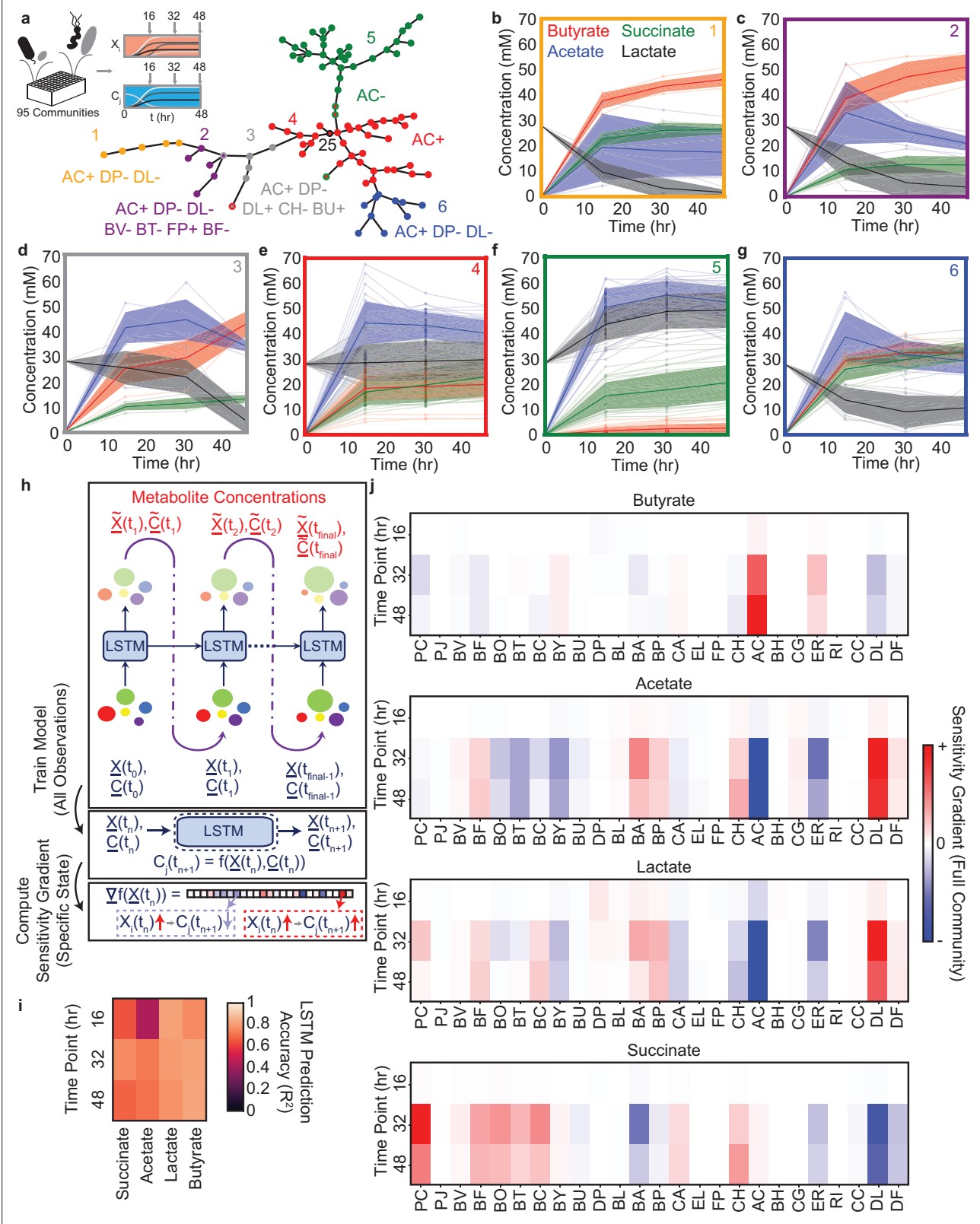

**Figure 5.** Community metabolite trajectories cluster into qualitatively distinct groups which can be classified based on presence and absence of key microbial species. (**a**) Schematic of experiment and network representing a minimal spanning tree across the 95 communities where weights (indicated by edge length) are equal to the Euclidean distance between the metabolite trajectories for each community. Node colors indicate clusters determined as described in the Materials and methods. Red node with black outline annotated with '25' represents the 25-member community. Annotations indicate

*Figure 5 continued on next page*

*Figure 5 continued*

the most specific microbial species presence/absence rules that describe most data points in the cluster of the corresponding color as determined by a decision tree classifier (Materials and methods). Communities that deviate from the rules for their cluster are indicated with a border matching the color of the closest cluster whose rules they do follow. Network visualization generated using the draw_kamada_kawai function in networkx (v2.1) for Python 3. (**b–g**) Temporal changes in metabolite concentrations for communities within each cluster (indicated by sub-plot border color), with individual communities denoted by transparent lines. Solid lines and shaded regions represent the mean ±1 s.d. of all communities in the cluster. (**h**) Schematic of LSTM model training and computation of gradients to evaluate impact of species abundance on metabolite concentrations in a specific community context. (**i**) Heatmap of model M3 prediction accuracy for four metabolites in the 34 validation communities at each time point (Pearson correlation $R^2$, N=34 for all tests). (**j**) Heatmap of the gradient analysis of model M3 as described in (**h**) for the full 25-species community. N and p-values are reported in *Supplementary file 1*.

The online version of this article includes the following figure supplement(s) for figure 5:

**Figure supplement 1.** Characteristics of the dynamic community behaviors.

**Figure supplement 2.** Prediction accuracy of model M3 for species abundance.

**Figure supplement 3.** Comparison of the discrete generalized Lotka-Volterra model to the LSTM using the same training algorithm.

a high fraction of 'distributed' communities (*Figure 3b*). Clusters with a smaller number of communities contained a higher percentage of 'corner' communities (*Figure 5—figure supplement 1b*,c). Therefore, the LSTM model informed by endpoint measurements of species abundance and metabolite concentrations elucidated 'corner' communities with qualitatively distinct temporal behaviors. These communities were unlikely to be discovered via random sampling of sub-communities due to the high density of points towards the center of the distribution and low density of communities in the tails of the distribution (*Figure 3b*). Additionally, some 'corner' communities that were similar in metabolite profiles when considering the endpoint measurement separated into different clusters when considering the dynamic data (e.g. Clusters 2 and 3, which have similar metabolite profiles at 48 hr but qualitatively distinct dynamics) (*Figure 5b*). This demonstrates that using a community design approach to explore the extremes of system behaviors with a limited time resolution enabled the identification of new features when the communities with extreme functions were characterized with higher time resolution.

To identify general patterns in species presence/absence of these communities that could explain the temporal behaviors of each cluster, we used a decision tree analysis to identify an interpretable classification scheme (*Figure 5—figure supplement 1d*). Using this approach, the large clusters were separated by relatively simple classification rules (i.e. AC+ for Cluster 4 and AC- for Cluster 5), whereas the smaller clusters had more complex classification rules involving larger combinations of species (3–7 species), all involving AC, DP, and DL (*Figure 5a*). The influential role of DP was corroborated by a previous study showing that DP substantially inhibits butyrate production (*Clark et al., 2021*). In addition, the inferred microbe-metabolite networks based on the LSTM model M2 demonstrated that the presence of DL was linked to higher acetate and lower succinate production (*Figure 3d*), consistent with its key role in shaping metabolite dynamics in this system. The variation in the number of communities across clusters is consistent with previous observations that species-rich microbial communities tend towards similar behavior(s) (e.g. Clusters 4 and 5 contained many communities). By contrast, more complex species presence/absence design rules are required to identify communities that deviate from this typical behavior (e.g. Clusters 1–3 and 6 contained few communities) (*Clark et al., 2021*).

## Using LSTM with higher time-resolution to interpret contextual interactions

While our clustering analysis identified general design rules for metabolite trajectories, there remained unexplained within-cluster variation. Thus, we used the LSTM framework to identify those effects beyond these general species presence/absence rules that determine the precise metabolite trajectory of a given community. Simultaneous predictions of species abundance and the concentration of all four metabolites at all time points necessitates specific modifications to the LSTM architecture shown in *Figure 2a*. In particular, we consider a 29-dimensional input vector whose first 25 components correspond to the species abundance, while the remaining four components correspond to the concentration of metabolites (*Figure 5h*). The 29-dimensional feature vector is suitably normalized so that the different components have zero mean and unity variance. The feature scaling is important to

prevent dominance of high-abundance species. The output of each LSTM unit is fed into the input block of the subsequent LSTM unit in order to advance the model forward in time. The reason behind concatenating instantaneous species abundances with metabolite concentrations can be understood as follows. Prediction of metabolite concentrations at various time points requires a time-series model (either using ODEs or LSTM in this case). Further, the future trajectory of metabolite concentrations is a function of both the species abundance, as well as the metabolite concentrations at the current time instant. Therefore, we concatenate both the metabolite concentrations and species abundances to create a 29-dimensional feature vector. The trained LSTM framework on the 60 training communities (model M3) displayed good prediction performance on the metabolite concentrations of the 34 validation communities plus the full 25-species community (*Figure 5i*). The prediction accuracy of species abundance was lower than metabolite concentrations, presumably due to the limited number of training set observations of each species (*Figure 5—figure supplement 2*).

We used a gradient-based sensitivity analysis of the LSTM model M3 to provide biological insights into the contributions of individual species based on the temporal changes in metabolite concentrations (*Figure 5h and j*, Methods). This method involves computing partial derivatives of output variables of interest with respect to input variables, which are readily available through a single backpropagation pass (*LeCun et al., 1988*; *Peurifoy et al., 2018*). As an example case, we applied this analysis approach to the full 25-species community, which was grouped into Cluster 4, with the design rule 'AC+' (*Figure 5a*). Consistent with this design rule, we observed strong sensitivity gradients between the abundance of AC and the concentrations of butyrate, acetate, and lactate, consistent with our biological understanding of the system (*Clark et al., 2021*). Beyond the 'AC+' design rule, there was a strong sensitivity gradient between DL and acetate and succinate, consistent with the inferred networks based on the LSTM model M2 that used endpoint measurements (*Figure 3d*). Further, the contributions of certain species on metabolite production varied as a function of time. For instance, in the initial time point, species abundances were similar and thus the contribution of individual species to metabolite production was uniform. However, interactions between species during community assembly enhanced the contribution of specific metabolite driver species such as AC. In addition, the contributions of individual species such as PC and BA to succinate production peaked at 32 hr and then decreased by 48 hr, highlighting that the effects of these species were maximized at intermediate time points. In sum, the gradient-based method identified the quantitative contributions of each species to the temporal changes in metabolite concentrations for a representative 25-member community, identifying context-specific behaviors beyond the previously identified broader design rules. These two complementary approaches are useful for identifying design rules governing metabolite dynamics. The clustering method can identify broad design rules for species presence/absence and the LSTM analysis gradient approach can uncover fine-tuned quantitative contributions of species to the temporal changes in community-level functions.

To directly evaluate the performance of the gLV and LSTM model, we trained a discretized version of the gLV model (approximate gLV model) on the same dataset and used the same algorithm as the LSTM. The approximate gLV model was augmented with a two layer feed-forward neural network with a hidden dimension equivalent to the hidden dimension used in the LSTM model to enable metabolite predictions (*Figure 5—figure supplement 3a, b*). The approximate gLV model enables the computation of gradients via the backpropagation algorithm, which is also used to train the LSTM. By contrast, computation of gradients of the continuous-time gLV model requires numerical integration. This approximate gLV model does not perform as well as the LSTM model at species abundance predictions using the same data used to train LSTM model M3 (*Figure 5—figure supplement 3c, b*). In addition, the LSTM outperforms the approximate gLV augmented with the feed-forward network at metabolite predictions (*Figure 5—figure supplement 3e, f*). In sum, the LSTM outperforms the discretized gLV model using the same training algorithm, highlighting the power of the LSTM model in accurately predicting the temporal changes in microbiome composition and metabolite concentrations.

## Discussion

The LSTM modeling framework trained on species abundance and metabolite concentrations accurately predicted multiple health-relevant functions of complex synthetic human gut communities. This model is powerful for designing communities with target metabolite profiles. Microbial communities

continuously impact metabolites by releasing or consuming them. Therefore, by modeling both microbial growth and the metabolites they produce/consume together, the LSTM captured the interconnections between these variables. Due to its flexibility, the LSTM model outperforms the widely used gLV model in the presence of higher-order interactions. We leveraged the computational efficiency of LSTM model to predict the metabolite profiles of tens of millions of communities. We used these model predictions to identify sparsely represented 'corner case' communities that maximized/minimized community-level production of four health-relevant metabolites. In the absence of a predictive model, these infrequent communities would have been difficult to discover among the vast metabolite profile landscape of possible communities.

Beyond the model's predictive capabilities, we showed that biological information including significant microbe-metabolite and microbe-microbe interactions, can be extracted from LSTM models. These biological insights could enable the discovery of key species and interactions driving community functions of interest. Further, this could inform the design of microbial communities from the bottom-up or interventions to manipulate community-level behaviors. For example, the inferred microbe-metabolite network highlighted AC is a major ecological driver of several metabolites including butyrate, acetate and lactate in our system. In addition, this microbe-metabolite network did not include species of the highly abundant genus *Bacteroides* but instead featured members of Firmicutes (AC, ER, DL), Actinobacteria (BA, BP, EL), Proteobacteria DP and Bacteroidetes PC. Notably, *Bacteroides* displayed numerous interactions in the microbe-microbe interaction network, suggesting that they played a key role in the growth of constituent community members opposed to production of specific metabolites. Therefore, our model suggests that *Bacteroides* influence broad ecosystem functions such as community growth dynamics whereas species highlighted in the microbe-metabolite network contribute to specialized functions such as the production of specific measured metabolites (*Rivett and Bell, 2018*). Therefore, the microbe-metabolite interaction network could be used to identify key species that could be targeted for manipulating the dynamics of specific metabolites.

We performed time-resolved measurements of metabolite production and species abundance using a set of designed communities and demonstrated that communities tend towards a typical dynamic behavior (i.e. Clusters 4 and 5). Therefore, random sampling of sub-communities from the 25-member system would likely exhibit behaviors similar to Clusters 4 and 5. We used the LSTM model to identify 'corner cases' communities that displayed metabolite concentrations near the tails of the metabolite distributions at the endpoint. The model allowed us to identify unique sub-clusters with disparate dynamic behaviors. We demonstrated that the endpoint model predictions were confirmatory (*Figure 3c*) and also led to new discoveries when additional measurements were made in the time dimension. Specifically, certain 'corner cases' communities identified based on prediction of a single time-point displayed distinct dynamic trajectories. For instance, Clusters 2 and 3 based on the decision tree classifier displayed similar end-point metabolite concentrations (*Figure 5c, d*). However, lactate decreased immediately over time in Cluster 2 communities but remained high until approximately 30 hr and then decreased in Cluster 3 communities. The design rule for Cluster 3 included the presence of lactate producers BU and DL (*Figure 3—figure supplement 4*), suggesting that these individual species' lactate producing capabilities enabled the community to maintain a high lactate concentration for an extended period of time in the context of the Cluster 3 communities. While we focused on the production of four health-relevant metabolites produced by gut microbiota, a wide range of health-relevant compounds are produced by gut bacteria. Therefore, communities that cluster together based on dynamic trends in the four measured metabolites could separate into new clusters based on the temporal patterns of other compounds produced or degraded by the communities.

Time-resolved measurements were required to reveal the different dynamic behaviors of communities in Clusters 2 and 3 to improve our understanding and the design of community functions. The ability to resolve differences in the dynamic trajectories of communities requires time sampling when the system behavior is changing as a function of time as opposed to time sampling once the system has reached a steady-state (i.e. saturated as a function of time). The time to reach steady-state varied across different communities and metabolites of interest. For instance, lactate reached steady-state at an earlier time point (12 hr) in Cluster 4 communities whereas communities in Cluster 3 approached steady-state at a later time point (48 hr). Therefore, model-guided experimental planning could be

used to identify the optimal sampling times to resolve differences in community dynamic behaviors. Achieving a highly predictive LSTM model required substantially less training data than a previous study that approximated the behavior of mechanistic biological systems models with RNNs (*Figure 2*; *Wang et al., 2019*). While the performance of any data-driven algorithm improves with the quantity and quality of available data, we demonstrate that the LSTM can translate learning on lower-order communities to accurately predict the behavior of higher-order communities given a limited and informative training set that is experimentally feasible. For synthetic microbial communities, the quality of the training set depends on the frequency of time-series measurements within periods in which the system displays rich dynamic behaviors (i.e. excitation of the dynamic modes of the system), the range of initial species richness, representation of each community member in the training data and sufficient variation in species abundances or metabolite concentrations (*Figure 4—figure supplement 1*). The dynamic behaviors of the synthetic communities characterized in vitro may likely exhibit significant differences to their behaviors in new environments such as the mammalian gut. However, communities in sub-clusters whose behaviors deviated substantially from the typical community behaviors (e.g. Clusters 2 and 3 versus Clusters 4 and 5) may be more likely than random to display unique dynamic behaviors in vivo. Future work will investigate whether the in vitro dynamic behavior cluster patterns can be used as prior information to guide the design of informative communities in new environments for building predictive models.

The current implementation of the LSTM model lacks uncertainty quantification for individual predictions, which could be used to guide experimental design (*Radivojević et al., 2020*). Recent progress in using Bayesian recurrent neural networks has led to emergence of Bayesian LSTMs (*Fortunato et al., 2017*; *Li et al., 2021*), which provides uncertainty quantification for each prediction in the form of posterior variance or posterior confidence interval. However, currently, the implementation and training of such Bayesian neural networks can be significantly more difficult than training the LSTM model developed here. In addition, we benchmarked the performance of the LSTM against a widely used gLV model which has been demonstrated to accurately predict community assembly in communities with up to 25 species (*Venturelli et al., 2018*; *Clark et al., 2021*). The gLV model has been modified mathematically to capture more complex system behaviors (*McPeek, 2017*). However, implementation of these gLV models to represent the behaviors of microbiomes with a large number of interacting species poses major computational challenges.

While our current approach treated microbiome species composition as the sole set of design variables in a constant environmental background, microbiomes in reality are impacted by differences in the physicochemical composition of their environment (*Thompson et al., 2017*). Given sufficient observations of community behavior under varied environmental contexts (e.g. presence/absence of certain nutrients), our LSTM approach could be further leveraged to design complementary species and environmental compositions for desired microbiome functional dynamics. Further, we can leverage the wealth of biological information stored in the sequenced genomes of the constituent organisms. Integrating methods such as genome scale models (*Magnúsdóttir et al., 2017*) with our LSTM framework could leverage genomic information to enable predictions when the genomes of the organisms are varied (i.e. alternative strains of the same species with disparate metabolic capabilities). In this case, introducing variables representing the presence/absence of specific metabolic reactions would potentially enable the model to predict the impact of a species with a varied set of metabolic reactions on a given set of functions without new experimental observations. Integrating this information into the model could thus enable a mapping between genome information and community-level functions.

While previous approaches have used machine learning methods to predict microbiome functions based on microbiome species composition (*Le et al., 2020*; *Larsen et al., 2012*; *Thompson et al., 2019*), our approach is a major step forward in predicting the future temporal trajectory of microbiome functions from an initial species composition. The dynamic nature of our approach enables the design of optimal initial community compositions or interventions to steer a community to a desired future state. The flexibility of our approach to various time resolutions could be especially useful in scenarios where a microbiome may display undesired transients on the path from an initial state to a desired final state. For instance, in treatment of gut microbiome dysbiosis, it is important to ensure that any transient states of the microbiome are not harmful to the host (e.g. pathogen blooms or overproduction of toxic metabolites) as the system approaches a desired healthy state (*Xiao et al., 2020*).

However, because predictions with increased time resolution require more data for model training, the ability of our approach to predict system behaviors based on initial and final observations is useful for scenarios where transient states may be less important, such as in bioprocesses where the concentration of products at the time of harvest is the primary design objective (*Yenkie et al., 2016*). Finally, the computational efficiency and accuracy of the LSTM model could be exploited in the future for autonomous design and optimization of multifunctional communities via computer-controlled design-test-learn cycles (*King et al., 2009*).

# Materials and methods

## Key resources table

| Reagent type (species) or resource | Designation | Source or reference | Identifiers | Additional information |
|---|---|---|---|---|
| Strain, strain background (*Prevotella copri* CB7) | PC | DSM 18205 | | |
| Strain, strain background (*Parabacteroides johnsonii* M-165) | PJ | DSM 18315 | | |
| Strain, strain background (*Bacteroides vulgatus* NCTC 11154) | BV | ATCC 8482 | | |
| Strain, strain background (*Bacteroides fragilis* EN-2) | BF | DSM 2151 | | |
| Strain, strain background (*Bacteroides ovatus* NCTC 11153) | BO | ATCC 8483 | | |
| Strain, strain background (*Bacteroides thetaiotaomicron* VPI 5482) | BT | ATCC 29148 | | |
| Strain, strain background (*Bacteroides caccae* VPI 3452 A) | BC | ATCC 43185 | | |
| Strain, strain background (*Bacteroides cellulosilyticus* CRE21) | BY | DSMZ 14838 | | |
| Strain, strain background (*Bacteroides uniformis* VPI 0061) | BU | DSM 6597 | | |
| Strain, strain background (*Desulfovibrio piger* VPI C3-23) | DP | ATCC 29098 | | |
| Strain, strain background (*Bifidobacterium longum* subs. infantis S12) | BL | DSM 20088 | | |
| Strain, strain background (*Bifidobacterium adolescentis* E194a (Variant a)) | BA | ATCC 15703 | | |
| Strain, strain background (*Bifidobacterium pseudocatenulatum* B1279) | BP | DSM 20438 | | |
| Strain, strain background (*Collinsella aerofaciens* VPI 1003) | CA | DSM 3979 | | |
| Strain, strain background (*Eggerthella lenta* 1899 B) | EL | DSM 2243 | | |
| Strain, strain background (*Faecalibacterium prausnitzii* A2-165) | FP | DSM 17677 | | |
| Strain, strain background (*Clostridium hiranonis* T0-931) | CH | DSM 13275 | | |
| Strain, strain background (*Anaerostipes caccae* L1-92) | AC | DSM 14662 | | |

*Continued on next page*

*Continued*

| Reagent type (species) or resource | Designation | Source or reference | Identifiers | Additional information |
|---|---|---|---|---|
| Strain, strain background (*Blautia hydrogenotrophica* S5a33) | BH | DSM 10507 | | |
| Strain, strain background (*Clostridium asparagiforme* N6) | CG | DSM 15981 | | |
| Strain, strain background (*Eubacterium rectale* VPI 0990) | ER | ATCC 33656 | | |
| Strain, strain background (*Roseburia intestinalis* L1-82) | RI | DSM 14610 | | |
| Strain, strain background (*Coprococcus comes* VPI CI-38) | CC | ATCC 27758 | | |
| Strain, strain background (*Dorea longicatena* 111–35) | DL | DSMZ 13814 | | |
| Strain, strain background (*Dorea formicigenerans* VPI C8-13) | DF | DSM 3992 | | |
| Sequence-based reagent | Forward Primer Index: ATCACG | IDT | | AATGATACGGCGACCACCGAGATCTACAC ATCACG ACACTCTTTCCCTACA CGACGCTCTTCCGATCT ACTCCTACGGGAGGCAGCAGT |
| Sequence-based reagent | Forward Primer Index: CGATGT | IDT | | AATGATACGGCGACCACCGAGATCTACAC CGATGT ACACTCTTTCCCTACA CGACGCTCTTCCGATCT T ACTCCTACGGGAGGCAGCAGT |
| Sequence-based reagent | Forward Primer Index: TTAGGC | IDT | | AATGATACGGCGACCACCGAGATCTACAC TTAGGC ACACTCTTTCCCTACA CGACGCTCTTCCGATCT GT ACTCCTACGGGAGGCAGCAGT |
| Sequence-based reagent | Forward Primer Index: TGACCA | IDT | | AATGATACGGCGACCACCGAGATCTACAC TGACCA ACACTCTTTCCCTACA CGACGCTCTTCCGATCT CGA ACTCCTACGGGAGGCAGCAGT |
| Sequence-based reagent | Forward Primer Index: ACAGTG | IDT | | AATGATACGGCGACCACCGAGATCTACAC ACAGTG ACACTCTTTCCCTACA CGACGCTCTTCCGATCT ATGA ACTCCTACGGGAGGCAGCAGT |
| Sequence-based reagent | Forward Primer Index: GCCAAT | IDT | | AATGATACGGCGACCACCGAGATCTACAC GCCAAT ACACTCTTTCCCTACA CGACGCTCTTCCGATCT TGCGA ACTCCTACGGGAGGCAGCAGT |
| Sequence-based reagent | Forward Primer Index: CAGATC | IDT | | AATGATACGGCGACCACCGAGATCTACAC CAGATC ACACTCTTTCCCTACA CGACGCTCTTCCGATCT GAGTGG ACTCCTACGGGAGGCAGCAGT |
| Sequence-based reagent | Forward Primer Index: ACTTGA | IDT | | AATGATACGGCGACCACCGAGATCTACAC ACTTGA ACACTCTTTCCCTACA CGACGCTCTTCCGATCT CCTGGAG ACTCCTACGGGAGGCAGCAGT |
| Sequence-based reagent | Forward Primer Index: GATCAG | IDT | | AATGATACGGCGACCACCGAGATCTACAC GATCAG ACACTCTTTCCCTACA CGACGCTCTTCCGATCT ACTCCTACGGGAGGCAGCAGT |
| Sequence-based reagent | Forward Primer Index: TAGCTT | IDT | | AATGATACGGCGACCACCGAGATCTACAC TAGCTT ACACTCTTTCCCTACA CGACGCTCTTCCGATCT T ACTCCTACGGGAGGCAGCAGT |
| Sequence-based reagent | Forward Primer Index: GGCTAC | IDT | | AATGATACGGCGACCACCGAGATCTACAC GGCTAC ACACTCTTTCCCTACA CGACGCTCTTCCGATCT GT ACTCCTACGGGAGGCAGCAGT |
| Sequence-based reagent | Forward Primer Index: CTTGTA | IDT | | AATGATACGGCGACCACCGAGATCTACAC CTTGTA ACACTCTTTCCCTACA CGACGCTCTTCCGATCT CGA ACTCCTACGGGAGGCAGCAGT |
| Sequence-based reagent | Forward Primer Index: AGTCAA | IDT | | AATGATACGGCGACCACCGAGATCTACAC AGTCAA ACACTCTTTCCCTACA CGACGCTCTTCCGATCT ATGA ACTCCTACGGGAGGCAGCAGT |
| Sequence-based reagent | Forward Primer Index: AGTTCC | IDT | | AATGATACGGCGACCACCGAGATCTACAC AGTTCC ACACTCTTTCCCTACA CGACGCTCTTCCGATCT TGCGA ACTCCTACGGGAGGCAGCAGT |
| Sequence-based reagent | Forward Primer Index: ATGTCA | IDT | | AATGATACGGCGACCACCGAGATCTACAC ATGTCA ACACTCTTTCCCTACA CGACGCTCTTCCGATCT GAGTGG ACTCCTACGGGAGGCAGCAGT |
| Sequence-based reagent | Forward Primer Index: CCGTCC | IDT | | AATGATACGGCGACCACCGAGATCTACAC CCGTCC ACACTCTTTCCCTACA CGACGCTCTTCCGATCT CCTGGAG ACTCCTACGGGAGGCAGCAGT |
| Sequence-based reagent | Forward Primer Index: GTAGAG | IDT | | AATGATACGGCGACCACCGAGATCTACAC GTAGAG ACACTCTTTCCCTACA CGACGCTCTTCCGATCT ACTCCTACGGGAGGCAGCAGT |
| Sequence-based reagent | Forward Primer Index: GTCCGC | IDT | | AATGATACGGCGACCACCGAGATCTACAC GTCCGC ACACTCTTTCCCTACA CGACGCTCTTCCGATCT T ACTCCTACGGGAGGCAGCAGT |

*Continued on next page*

*Continued*

| Reagent type (species) or resource | Designation | Source or reference | Identifiers | Additional information |
|---|---|---|---|---|
| Sequence-based reagent | Forward Primer Index: GTGAAA | IDT | | AATGATACGGCGACCACCGAGATCTACAC GTGAAA ACACTCTTTCCCTACA CGACGCTCTTCCGATCT GT ACTCCTACGGGAGGCAGCAGT |
| Sequence-based reagent | Forward Primer Index: GTGGCC | IDT | | AATGATACGGCGACCACCGAGATCTACAC GTGGCC ACACTCTTTCCCTACA CGACGCTCTTCCGATCT CGA ACTCCTACGGGAGGCAGCAGT |
| Sequence-based reagent | Forward Primer Index: GTTTCG | IDT | | AATGATACGGCGACCACCGAGATCTACAC GTTTCG ACACTCTTTCCCTACA CGACGCTCTTCCGATCT ATGA ACTCCTACGGGAGGCAGCAGT |
| Sequence-based reagent | Forward Primer Index: CGTACG | IDT | | AATGATACGGCGACCACCGAGATCTACAC CGTACG ACACTCTTTCCCTACA CGACGCTCTTCCGATCT TGCGA ACTCCTACGGGAGGCAGCAGT |
| Sequence-based reagent | Forward Primer Index: GAGTGG | IDT | | AATGATACGGCGACCACCGAGATCTACAC GAGTGG ACACTCTTTCCCTACA CGACGCTCTTCCGATCT GAGTGG ACTCCTACGGGAGGCAGCAGT |
| Sequence-based reagent | Forward Primer Index: GGTAGC | IDT | | AATGATACGGCGACCACCGAGATCTACAC GGTAGC ACACTCTTTCCCTACA CGACGCTCTTCCGATCT CCTGGAG ACTCCTACGGGAGGCAGCAGT |
| Sequence-based reagent | Forward Primer Index: ACTGAT | IDT | | AATGATACGGCGACCACCGAGATCTACAC ACTGAT ACACTCTTTCCCTACA CGACGCTCTTCCGATCT ACTCCTACGGGAGGCAGCAGT |
| Sequence-based reagent | Forward Primer Index: ATGAGC | IDT | | AATGATACGGCGACCACCGAGATCTACAC ATGAGC ACACTCTTTCCCTACA CGACGCTCTTCCGATCT T ACTCCTACGGGAGGCAGCAGT |
| Sequence-based reagent | Forward Primer Index: ATTCCT | IDT | | AATGATACGGCGACCACCGAGATCTACAC ATTCCT ACACTCTTTCCCTACA CGACGCTCTTCCGATCT GT ACTCCTACGGGAGGCAGCAGT |
| Sequence-based reagent | Forward Primer Index: CAAAAG | IDT | | AATGATACGGCGACCACCGAGATCTACAC CAAAAG ACACTCTTTCCCTACA CGACGCTCTTCCGATCT CGA ACTCCTACGGGAGGCAGCAGT |
| Sequence-based reagent | Forward Primer Index: CAACTA | IDT | | AATGATACGGCGACCACCGAGATCTACAC CAACTA ACACTCTTTCCCTACA CGACGCTCTTCCGATCT ATGA ACTCCTACGGGAGGCAGCAGT |
| Sequence-based reagent | Forward Primer Index: CACCGG | IDT | | AATGATACGGCGACCACCGAGATCTACAC CACCGG ACACTCTTTCCCTACA CGACGCTCTTCCGATCT TGCGA ACTCCTACGGGAGGCAGCAGT |
| Sequence-based reagent | Forward Primer Index: CACGAT | IDT | | AATGATACGGCGACCACCGAGATCTACAC CACGAT ACACTCTTTCCCTACA CGACGCTCTTCCGATCT GAGTGG ACTCCTACGGGAGGCAGCAGT |
| Sequence-based reagent | Forward Primer Index: CACTCA | IDT | | AATGATACGGCGACCACCGAGATCTACAC CACTCA ACACTCTTTCCCTACA CGACGCTCTTCCGATCT CCTGGAG ACTCCTACGGGAGGCAGCAGT |
| Sequence-based reagent | Forward Primer Index: CAGGCG | IDT | | AATGATACGGCGACCACCGAGATCTACAC CAGGCG ACACTCTTTCCCTACA CGACGCTCTTCCGATCT ACTCCTACGGGAGGCAGCAGT |
| Sequence-based reagent | Forward Primer Index: CATGGC | IDT | | AATGATACGGCGACCACCGAGATCTACAC CATGGC ACACTCTTTCCCTACA CGACGCTCTTCCGATCT T ACTCCTACGGGAGGCAGCAGT |
| Sequence-based reagent | Forward Primer Index: CATTTT | IDT | | AATGATACGGCGACCACCGAGATCTACAC CATTTT ACACTCTTTCCCTACA CGACGCTCTTCCGATCT GT ACTCCTACGGGAGGCAGCAGT |
| Sequence-based reagent | Forward Primer Index: CCAACA | IDT | | AATGATACGGCGACCACCGAGATCTACAC CCAACA ACACTCTTTCCCTACA CGACGCTCTTCCGATCT CGA ACTCCTACGGGAGGCAGCAGT |
| Sequence-based reagent | Forward Primer Index: CGGAAT | IDT | | AATGATACGGCGACCACCGAGATCTACAC CGGAAT ACACTCTTTCCCTACA CGACGCTCTTCCGATCT ATGA ACTCCTACGGGAGGCAGCAGT |
| Sequence-based reagent | Forward Primer Index: CTAGCT | IDT | | AATGATACGGCGACCACCGAGATCTACAC CTAGCT ACACTCTTTCCCTACA CGACGCTCTTCCGATCT TGCGA ACTCCTACGGGAGGCAGCAGT |
| Sequence-based reagent | Forward Primer Index: CTATAC | IDT | | AATGATACGGCGACCACCGAGATCTACAC CTATAC ACACTCTTTCCCTACA CGACGCTCTTCCGATCT GAGTGG ACTCCTACGGGAGGCAGCAGT |
| Sequence-based reagent | Forward Primer Index: CTCAGA | IDT | | AATGATACGGCGACCACCGAGATCTACAC CTCAGA ACACTCTTTCCCTACA CGACGCTCTTCCGATCT CCTGGAG ACTCCTACGGGAGGCAGCAGT |
| Sequence-based reagent | Forward Primer Index: GACGAC | IDT | | AATGATACGGCGACCACCGAGATCTACAC GACGAC ACACTCTTTCCCTACA CGACGCTCTTCCGATCT ACTCCTACGGGAGGCAGCAGT |
| Sequence-based reagent | Forward Primer Index: TAATCG | IDT | | AATGATACGGCGACCACCGAGATCTACAC TAATCG ACACTCTTTCCCTACA CGACGCTCTTCCGATCT T ACTCCTACGGGAGGCAGCAGT |
| Sequence-based reagent | Forward Primer Index: TACAGC | IDT | | AATGATACGGCGACCACCGAGATCTACAC TACAGC ACACTCTTTCCCTACA CGACGCTCTTCCGATCT GT ACTCCTACGGGAGGCAGCAGT |

*Continued on next page*

*Continued*

| Reagent type (species) or resource | Designation | Source or reference | Identifiers | Additional information |
|---|---|---|---|---|
| Sequence-based reagent | Forward Primer Index: TATAAT | IDT | | AATGATACGGCGACCACCGAGATCTACAC TATAAT ACACTCTTTCCCTACA CGACGCTCTTCCGATCT CGA ACTCCTACGGGAGGCAGCAGT |
| Sequence-based reagent | Forward Primer Index: TCATTC | IDT | | AATGATACGGCGACCACCGAGATCTACAC TCATTC ACACTCTTTCCCTACA CGACGCTCTTCCGATCT ATGA ACTCCTACGGGAGGCAGCAGT |
| Sequence-based reagent | Forward Primer Index: TCCCGA | IDT | | AATGATACGGCGACCACCGAGATCTACAC TCCCGA ACACTCTTTCCCTACA CGACGCTCTTCCGATCT TGCGA ACTCCTACGGGAGGCAGCAGT |
| Sequence-based reagent | Forward Primer Index: TCGAAG | IDT | | AATGATACGGCGACCACCGAGATCTACAC TCGAAG ACACTCTTTCCCTACA CGACGCTCTTCCGATCT GAGTGG ACTCCTACGGGAGGCAGCAGT |
| Sequence-based reagent | Forward Primer Index: TCGGCA | IDT | | AATGATACGGCGACCACCGAGATCTACAC TCGGCA ACACTCTTTCCCTACA CGACGCTCTTCCGATCT CCTGGAG ACTCCTACGGGAGGCAGCAGT |
| Sequence-based reagent | Reverse Primer Index: ATCACGAG | IDT | | CAAGCAGAAGACGGCATACGAGAT ATCACGAG GTGACTGGAGTTCAGA CGTGTGCTCTTCCGATCT ggactaccagggtatctaatcctgt |
| Sequence-based reagent | Reverse Primer Index: CGATGTTC | IDT | | CAAGCAGAAGACGGCATACGAGAT CGATGTTC GTGACTGGAGTTCAGA CGTGTGCTCTTCCGATCT A ggactaccagggtatctaatcctgt |
| Sequence-based reagent | Reverse Primer Index: TTAGGCGA | IDT | | CAAGCAGAAGACGGCATACGAGAT TTAGGCGA GTGACTGGAGTTCAGA CGTGTGCTCTTCCGATCT TC ggactaccagggtatctaatcctgt |
| Sequence-based reagent | Reverse Primer Index: TGACCAAT | IDT | | CAAGCAGAAGACGGCATACGAGAT TGACCAAT GTGACTGGAGTTCAGA CGTGTGCTCTTCCGATCT CTA ggactaccagggtatctaatcctgt |
| Sequence-based reagent | Reverse Primer Index: ACAGTGCT | IDT | | CAAGCAGAAGACGGCATACGAGAT ACAGTGCT GTGACTGGAGTTCAGA CGTGTGCTCTTCCGATCT GATA ggactaccagggtatctaatcctgt |
| Sequence-based reagent | Reverse Primer Index: GCCAATGT | IDT | | CAAGCAGAAGACGGCATACGAGAT GCCAATGT GTGACTGGAGTTCAGA CGTGTGCTCTTCCGATCT ACTCA ggactaccagggtatctaatcctgt |
| Sequence-based reagent | Reverse Primer Index: CAGATCGA | IDT | | CAAGCAGAAGACGGCATACGAGAT CAGATCGA GTGACTGGAGTTCAGA CGTGTGCTCTTCCGATCT TTCTCT ggactaccagggtatctaatcctgt |
| Sequence-based reagent | Reverse Primer Index: ACTTGAAA | IDT | | CAAGCAGAAGACGGCATACGAGAT ACTTGAAA GTGACTGGAGTTCAGA CGTGTGCTCTTCCGATCT CACTTCT ggactaccagggtatctaatcctgt |
| Sequence-based reagent | Reverse Primer Index: GATCAGTG | IDT | | CAAGCAGAAGACGGCATACGAGAT GATCAGTG GTGACTGGAGTTCAGA CGTGTGCTCTTCCGATCT ggactaccagggtatctaatcctgt |
| Sequence-based reagent | Reverse Primer Index: TCTACCTC | IDT | | CAAGCAGAAGACGGCATACGAGAT TCTACCTC GTGACTGGAGTTCAGA CGTGTGCTCTTCCGATCT A ggactaccagggtatctaatcctgt |
| Sequence-based reagent | Reverse Primer Index: CTTGTATG | IDT | | CAAGCAGAAGACGGCATACGAGAT CTTGTATG GTGACTGGAGTTCAGA CGTGTGCTCTTCCGATCT TC ggactaccagggtatctaatcctgt |
| Sequence-based reagent | Reverse Primer Index: TAGCTTCC | IDT | | CAAGCAGAAGACGGCATACGAGAT TAGCTTCC GTGACTGGAGTTCAGA CGTGTGCTCTTCCGATCT CTA ggactaccagggtatctaatcctgt |
| Sequence-based reagent | Reverse Primer Index: GGCTACCA | IDT | | CAAGCAGAAGACGGCATACGAGAT GGCTACCA GTGACTGGAGTTCAGA CGTGTGCTCTTCCGATCT GATA ggactaccagggtatctaatcctgt |
| Sequence-based reagent | Reverse Primer Index: ATGCACTT | IDT | | CAAGCAGAAGACGGCATACGAGAT ATGCACTT GTGACTGGAGTTCAGA CGTGTGCTCTTCCGATCT ACTCA ggactaccagggtatctaatcctgt |
| Sequence-based reagent | Reverse Primer Index: GACGGAAC | IDT | | CAAGCAGAAGACGGCATACGAGAT GACGGAAC GTGACTGGAGTTCAGA CGTGTGCTCTTCCGATCT TTCTCT ggactaccagggtatctaatcctgt |
| Sequence-based reagent | Reverse Primer Index: AGCCTTGG | IDT | | CAAGCAGAAGACGGCATACGAGAT AGCCTTGG GTGACTGGAGTTCAGA CGTGTGCTCTTCCGATCT CACTTCT ggactaccagggtatctaatcctgt |
| Sequence-based reagent | Reverse Primer Index: CCGTAGAG | IDT | | CAAGCAGAAGACGGCATACGAGAT CCGTAGAG GTGACTGGAGTTCAGA CGTGTGCTCTTCCGATCT ggactaccagggtatctaatcctgt |
| Sequence-based reagent | Reverse Primer Index: GTGAGACT | IDT | | CAAGCAGAAGACGGCATACGAGAT GTGAGACT GTGACTGGAGTTCAGA CGTGTGCTCTTCCGATCT A ggactaccagggtatctaatcctgt |
| Sequence-based reagent | Reverse Primer Index: AATGCTCA | IDT | | CAAGCAGAAGACGGCATACGAGAT AATGCTCA GTGACTGGAGTTCAGA CGTGTGCTCTTCCGATCT TC ggactaccagggtatctaatcctgt |
| Sequence-based reagent | Reverse Primer Index: GCATCGTA | IDT | | CAAGCAGAAGACGGCATACGAGAT GCATCGTA GTGACTGGAGTTCAGA CGTGTGCTCTTCCGATCT CTA ggactaccagggtatctaatcctgt |

*Continued on next page*

*Continued*

| Reagent type (species) or resource | Designation | Source or reference | Identifiers | Additional information |
|---|---|---|---|---|
| Sequence-based reagent | Reverse Primer Index: CGAACAGC | IDT | | CAAGCAGAAGACGGCATACGAGAT CGAACAGC GTGACTGGAGTTCAGA CGTGTGCTCTTCCGATCT GATA ggactaccagggtatctaatcctgt |
| Sequence-based reagent | Reverse Primer Index: TCGGAAGG | IDT | | CAAGCAGAAGACGGCATACGAGAT TCGGAAGG GTGACTGGAGTTCAGA CGTGTGCTCTTCCGATCT ACTCA ggactaccagggtatctaatcctgt |
| Sequence-based reagent | Reverse Primer Index: TTCTGTCG | IDT | | CAAGCAGAAGACGGCATACGAGAT TTCTGTCG GTGACTGGAGTTCAGA CGTGTGCTCTTCCGATCT TTCTCT ggactaccagggtatctaatcctgt |
| Sequence-based reagent | Reverse Primer Index: GTACTCAC | IDT | | CAAGCAGAAGACGGCATACGAGAT GTACTCAC GTGACTGGAGTTCAGA CGTGTGCTCTTCCGATCT CACTTCT ggactaccagggtatctaatcctgt |
| Sequence-based reagent | Reverse Primer Index: AGTAATAC | IDT | | CAAGCAGAAGACGGCATACGAGAT AGTAATAC GTGACTGGAGTTCAGA CGTGTGCTCTTCCGATCT ggactaccagggtatctaatcctgt |
| Sequence-based reagent | Reverse Primer Index: CAAGATAT | IDT | | CAAGCAGAAGACGGCATACGAGAT CAAGATAT GTGACTGGAGTTCAGA CGTGTGCTCTTCCGATCT A ggactaccagggtatctaatcctgt |
| Sequence-based reagent | Reverse Primer Index: TGTTTGGT | IDT | | CAAGCAGAAGACGGCATACGAGAT TGTTTGGT GTGACTGGAGTTCAGA CGTGTGCTCTTCCGATCT TC ggactaccagggtatctaatcctgt |
| Sequence-based reagent | Reverse Primer Index: CTCCAACC | IDT | | CAAGCAGAAGACGGCATACGAGAT CTCCAACC GTGACTGGAGTTCAGA CGTGTGCTCTTCCGATCT CTA ggactaccagggtatctaatcctgt |
| Sequence-based reagent | Reverse Primer Index: AAATTCTG | IDT | | CAAGCAGAAGACGGCATACGAGAT AAATTCTG GTGACTGGAGTTCAGA CGTGTGCTCTTCCGATCT GATA ggactaccagggtatctaatcctgt |
| Sequence-based reagent | Reverse Primer Index: CCCGCCAA | IDT | | CAAGCAGAAGACGGCATACGAGAT CCCGCCAA GTGACTGGAGTTCAGA CGTGTGCTCTTCCGATCT ACTCA ggactaccagggtatctaatcctgt |
| Sequence-based reagent | Reverse Primer Index: TACAAATA | IDT | | CAAGCAGAAGACGGCATACGAGAT TACAAATA GTGACTGGAGTTCAGACGTG TGCTCTTCCGATCT TTCTCT ggactaccagggtatctaatcctgt |
| Sequence-based reagent | Reverse Primer Index: GGGCTATA | IDT | | CAAGCAGAAGACGGCATACGAGAT GGGCTATA GTGACTGGAGTTCAGA CGTGTGCTCTTCCGATCT CACTTCT ggactaccagggtatctaatcctgt |
| Sequence-based reagent | Reverse Primer Index: TTTCGGAC | IDT | | CAAGCAGAAGACGGCATACGAGAT TTTCGGAC GTGACTGGAGTTCAGA CGTGTGCTCTTCCGATCT ggactaccagggtatctaatcctgt |
| Sequence-based reagent | Reverse Primer Index: TGCGCGTC | IDT | | CAAGCAGAAGACGGCATACGAGAT TGCGCGTC GTGACTGGAGTTCAGA CGTGTGCTCTTCCGATCT A ggactaccagggtatctaatcctgt |
| Sequence-based reagent | Reverse Primer Index: TCCCGCTG | IDT | | CAAGCAGAAGACGGCATACGAGAT TCCCGCTG GTGACTGGAGTTCAGA CGTGTGCTCTTCCGATCT TC ggactaccagggtatctaatcctgt |
| Sequence-based reagent | Reverse Primer Index: GTTTCAGG | IDT | | CAAGCAGAAGACGGCATACGAGAT GTTTCAGG GTGACTGGAGTTCAGA CGTGTGCTCTTCCGATCT CTA ggactaccagggtatctaatcctgt |
| Sequence-based reagent | Reverse Primer Index: GGAGGGGG | IDT | | CAAGCAGAAGACGGCATACGAGAT GGAGGGGG GTGACTGGAGTTCAGA CGTGTGCTCTTCCGATCT GATA ggactaccagggtatctaatcctgt |
| Sequence-based reagent | Reverse Primer Index: GCTGTTAG | IDT | | CAAGCAGAAGACGGCATACGAGAT GCTGTTAG GTGACTGGAGTTCAGA CGTGTGCTCTTCCGATCT ACTCA ggactaccagggtatctaatcctgt |
| Sequence-based reagent | Reverse Primer Index: GAGTGTGA | IDT | | CAAGCAGAAGACGGCATACGAGAT GAGTGTGA GTGACTGGAGTTCAGA CGTGTGCTCTTCCGATCT TTCTCT ggactaccagggtatctaatcctgt |
| Sequence-based reagent | Reverse Primer Index: CGTCCCCG | IDT | | CAAGCAGAAGACGGCATACGAGAT CGTCCCCG GTGACTGGAGTTCAGA CGTGTGCTCTTCCGATCT CACTTCT ggactaccagggtatctaatcctgt |
| Sequence-based reagent | Reverse Primer Index: CCTCATCA | IDT | | CAAGCAGAAGACGGCATACGAGAT CCTCATCA GTGACTGGAGTTCAGA CGTGTGCTCTTCCGATCT ggactaccagggtatctaatcctgt |
| Sequence-based reagent | Reverse Primer Index: CCACGACA | IDT | | CAAGCAGAAGACGGCATACGAGAT CCACGACA GTGACTGGAGTTCAGA CGTGTGCTCTTCCGATCT A ggactaccagggtatctaatcctgt |
| Sequence-based reagent | Reverse Primer Index: CATTGGCT | IDT | | CAAGCAGAAGACGGCATACGAGAT CATTGGCT GTGACTGGAGTTCAGA CGTGTGCTCTTCCGATCT TC ggactaccagggtatctaatcctgt |
| Sequence-based reagent | Reverse Primer Index: AGGGGCCC | IDT | | CAAGCAGAAGACGGCATACGAGAT AGGGGCCC GTGACTGGAGTTCAGA CGTGTGCTCTTCCGATCT CTA ggactaccagggtatctaatcctgt |
| Sequence-based reagent | Reverse Primer Index: ACGACACT | IDT | | CAAGCAGAAGACGGCATACGAGAT ACGACACT GTGACTGGAGTTCAGA CGTGTGCTCTTCCGATCT GATA ggactaccagggtatctaatcctgt |

*Continued on next page*

*Continued*

| Reagent type (species) or resource | Designation | Source or reference | Identifiers | Additional information |
|---|---|---|---|---|
| Sequence-based reagent | Reverse Primer Index: ACCGACGC | IDT | | CAAGCAGAAGACGGCATACGAGAT ACCGACGC GTGACTGGAGTTCAGA CGTGTGCTCTTCCGATCT ACTCA ggactaccagggtatctaatcctgt |
| Sequence-based reagent | Reverse Primer Index: TATAGTAT | IDT | | CAAGCAGAAGACGGCATACGAGAT TATAGTAT GTGACTGGAGTTCAGACGTG TGCTCTTCCGATCT TTCTCT ggactaccagggtatctaatcctgt |
| Sequence-based reagent | Reverse Primer Index: AACTCAGT | IDT | | CAAGCAGAAGACGGCATACGAGAT AACTCAGT GTGACTGGAGTTCAGA CGTGTGCTCTTCCGATCT CACTTCT ggactaccagggtatctaatcctgt |

## Experimental methods

### Strain maintenance and culturing

All anaerobic culturing was carried out in an anaerobic chamber with an atmosphere of 2.5 ± 0.5% $H_2$, 15±1% $CO_2$ and balance $N_2$. All prepared media and materials were placed in the chamber at least overnight before use to equilibrate with the chamber atmosphere. The strains used in this work were obtained from the sources listed in our previous publication (*Clark et al., 2021*) and permanent stocks of each were stored in 25% glycerol at -80°C. Batches of single-use glycerol stocks were produced for each strain by first growing a culture from the permanent stock in anaerobic basal broth (ABB) media (Oxoid) to stationary phase, mixing the culture in an equal volume of 50% glycerol, and aliquoting 400µL into Matrix Tubes (ThermoFisher) for storage at -80°C. Quality control for each batch of single-use glycerol stocks included (1) plating a sample of the aliquoted mixture onto LB media (Sigma-Aldrich) for incubation at 37°C in ambient air to detect aerobic contaminants and (2) Illumina sequencing of 16 S rDNA isolated from pellets of the aliquoted mixture to verify the identity of the organism. For each experiment, precultures of each species were prepared by thawing a single-use glycerol stock and combining the inoculation volume and media as described in *Clark et al., 2021* to a total volume of 5 mL (multiple tubes inoculated if more preculture volume needed). Cultures were incubated until stationary phase at 37°C using the preculture incubation times described in *Clark et al., 2021*. All experiments were performed in a chemically defined medium (DM38), as previously described (*Clark et al., 2021*). This medium supports the individual growth of all organisms except *Faecalibacterium prausnitzii* (*Clark et al., 2021*).

### Community culturing experiments and sample collection

Synthetic communities were assembled using liquid handling-based automation as described previously (*Clark et al., 2021*). Briefly, each species' preculture was diluted to an $OD_{600}$ of 0.0066 in DM38. Community combinations were arrayed in 96 deep well (96DW) plates by pipetting equal volumes of each species' diluted preculture into the appropriate wells using a Tecan Evo Liquid Handling Robot inside an anaerobic chamber. For experiments with multiple time points, duplicate 96DW plates were prepared for each time point. Each 96DW plate was covered with a semi-permeable membrane (Diversified Biotech) and incubated at 37°C. After the specified time had passed, 96DW plates were removed from the incubator and samples were mixed by pipette. Cell density was measured by pipetting 200µL of each sample into one 96-well microplate (96 W MP) and diluting 20 L of each sample into 180µL of PBS in another 96 W MP and measuring the $OD_{600}$ of both plates (Tecan F200 Plate Reader). We selected the value that was within the linear range of the instrument for each sample. A total of 200µL of each sample was transferred to a new 96DW plate and pelleted by centrifugation at 2400xg for 10 min. A supernatant volume of 180µL was removed from each sample and transferred to a 96-well microplate for storage at -20°C and subsequent metabolite quantification by high performance liquid chromatography (HPLC). Cell pellets were stored at -80°C for subsequent genomic DNA extraction and 16 S rDNA library preparation for Illumina sequencing. 20µL of each supernatant was used to quantify pH using a phenol Red assay (*Silverstein, 2012*). Phenol red solution was diluted to 0.05% weight per volume in 0.9% w/v NaCl. Bacterial supernatant (20µL) was added to 180µ of phenol red solution in a 96 W MP, and absorbance was measured at 560 nm (Tecan Spark Plate Reader). A standard curve was produced by fitting the Henderson-Hasselbach equation to fresh media with a pH ranging between 3 and 11 measured using a standard electro-chemical pH probe (Mettler-Toledo). We used (1) to map the pH values to the absorbance measurements.

$$pH = pK_a + b \cdot \log_{10}\left(\frac{A - A_{\min}}{A_{\max} - A}\right) \qquad (1)$$

The parameters $b$ and p$K_a$ were determined using a linear regression between pH and the log term for the standards in the linear range of absorbance (pH between 5.2 and 11) with $A_{\max}$ representing the absorbance of the pH 11 standard, $A_{\min}$ denoting the absorbance of the pH 3 standard and $A$ representing the absorbance of each condition.

## HPLC quantification of organic acids

Butyrate, succinate, lactate, and acetate concentrations in culture supernatants were quantified as described previously (*Clark et al., 2021*). Supernatant samples were thawed in a room temperature water bath before addition of 2μL of $H_2SO_4$ to precipitate any components that might be incompatible with the running buffer. The samples were then centrifuged at 2400xg for 10 min and then 150μL of each sample was filtered through a 0.2μm filter using a vacuum manifold before transferring 70μL of each sample to an HPLC vial. HPLC analysis was performed using a Shimadzu HPLC system equipped with a SPD-20AV UV detector (210 nm). Compounds were separated on a 250×4.6 mm Rezex OA-Organic acid LC column (Phenomenex Torrance, CA) run with a flow rate of 0.2 ml min$_{-1}$ and at a column temperature of -50°C. The samples were held at 4°C prior to injection. Separation was isocratic with a mobile phase of HPLC grade water acidified with 0.015 N $H_2SO_4$ ($415 \mu L L^{-1}$). At least two standard sets were run along with each sample set. Standards were 100, 20, and 4 mM concentrations of butyrate, succinate, lactate, and acetate, respectively. The injection volume for both sample and standard was 25μL. The resultant data was analyzed using the Shimadzu LabSolutions software package.

## Genomic DNA extraction and sequencing library preparation

Genomic DNA extraction and sequencing library preparation were performed as described previously (*Clark et al., 2021*). Genomic DNA was extracted from cell pellets using a modified version of the Qiagen DNeasy Blood and Tissue Kit protocol. First, pellets in 96DW plates were removed from -80°C and thawed in a room temperature water bath. Each pellet was resuspended in $180 \mu L$ of enzymatic lysis buffer (20 mM Tris-HCl (Invitrogen), 2 mM Sodium EDTA (Sigma-Aldrich), 1.2% Triton X-100 (Sigma-Aldrich), 20 mg/mL Lysozyme from chicken egg white (Sigma-Aldrich)). Plates were then covered with a foil seal and incubated at 37°C for 30 min with orbital shaking at 600 RPM. Then, $25 \mu L$ of $20 mg mL^{-1}$ Proteinase K (VWR) and 200 L of Buffer AL (QIAGEN) were added to each sample before mixing with a pipette. Plates were then covered by a foil seal and incubated at 56°C for 30 min with orbital shaking at 600 RPM. Next, $200 \mu L$ of 100% ethanol (Koptec) was added to each sample before mixing and samples were transferred to a Nucleic Acid Binding (NAB) plate (Pall) on a vacuum manifold with a 96DW collection plate. Each well in the NAB plate was then washed once with $500 \mu L$ Buffer AW1 (QIAGEN) and once with $500 \mu L$ of Buffer AW2 (QIAGEN). A vacuum was applied to the Pall NAB plate for an additional 10 min to remove any excess ethanol. Samples were then eluted into a clean 96DW plate from each well using $110 \mu L$ of Buffer AE (QIAGEN) preheated to 56°C. Genomic DNA samples were stored at -20°C until further processing.

Genomic DNA concentrations were measured using a SYBR Green fluorescence assay and then normalized to a concentration of $1 ng L^{-1}$ by diluting in molecular grade water using a Tecan Evo Liquid Handling Robot. First, genomic DNA samples were removed from -20°C and thawed in a room temperature water bath. Then, $1 \mu L$ of each sample was combined with $95 \mu L$ of SYBR Green (Invitrogen) diluted by a factor of 100 in TE Buffer (Integrated DNA Technologies) in a black 384-well microplate. This process was repeated with two replicates of each DNA standard with concentrations of 0, 0.5, 1, 2, 4, and $6 ng L^{-1}$. Each sample was then measured for fluorescence with an excitation/emission of 485/535 nm using a Tecan Spark plate reader. Concentrations of each sample were calculated using the standard curve and a custom Python script was used to compute the dilution factors and write a worklist for the Tecan Evo Liquid Handling Robot to normalize each sample to $1 ng L^{-1}$ in molecular grade water. Samples with DNA concentration less than $1 ng L^{-1}$ were not diluted. Diluted genomic DNA samples were stored at -20°C until further processing.

Amplicon libraries were generated from diluted genomic DNA samples by PCR amplification of the V3-V4 of the 16 S rRNA gene using custom dual-indexed primers for multiplexed next generation amplicon sequencing on Illumina platforms (*Clark et al., 2021*). Primers were arrayed in skirted 96-well PCR plates (VWR) using an acoustic liquid handling robot (Labcyte Echo 550) such that each

well received a different combination of one forward and one reverse primer ($0.1\mu L$ of each). After liquid evaporated, dry primers were stored at -20°C. Primers were resuspended in $15\mu L$ PCR master mix ($0.2\mu L$ Phusion High Fidelity DNA Polymerase [Thermo Scientific], $0.4\mu L$ 10 mM dNTP Solution [New England Biolabs], $4\mu L$ 5 x Phusion HF Buffer [Thermo Scientific], $4\mu L$ 5 M Betaine [Sigma-Aldrich], $6.4\mu L$ Water) and $5\mu L$ of normalized genomic DNA to give a final concentration of 0.05 M of each primer. Primer plates were sealed with Microplate B seals (Bio-Rad) and PCR was performed using a Bio-Rad C1000 Thermal Cycler with the following program: initial denaturation at 98°C (30 s); 25 cycles of denaturation at 98°C (10 s), annealing at 60°C (30 s), extension at 72°C (60 s); and final extension at 72°C (10 min). Of PCR products from each well, 2 µL were pooled and purified using the DNA Clean & Concentrator (Zymo) and eluted in water. The resulting libraries were sequenced on an Illumina MiSeq using a MiSeq Reagent Kit v3 (600-cycle) to generate 2 × 300 paired end reads.

### Bioinformatic analysis for quantification of species abundance

Sequencing data were used to quantify species relative abundance as described previously (*Clark et al., 2021*). Sequencing data were demultiplexed using Basespace Sequencing Hub's FastQ Generation program. Custom python scripts were used for further data processing as described previously (*Clark et al., 2021*). Paired end reads were merged using PEAR (v0.9.10) (*Zhang et al., 2014*) after which reads without forward and reverse annealing regions were filtered out. A reference database of the V3-V5 16 S rRNA gene sequences was created using consensus sequences from next-generation sequencing data or Sanger sequencing data of monospecies cultures. Sequences were mapped to the reference database using the mothur (v1.40.5) (*Schloss et al., 2009*) command classify.seqs (Wang method with a bootstrap cutoff value of 60). Relative abundance was calculated as the read count mapped to each species divided by the total number of reads for each condition. Absolute abundance of each species was calculated by multiplying the relative abundance by the $OD6^{00}$ measurement for each sample. Samples were excluded from further analysis if .1% of the reads were assigned to a species not expected to be in the community (indicating contamination). We expect the precision of our measurements to drop rapidly for species representing <1% of the community due to limited sequencing depth. We typically sequenced on the order of 10,000 molecules of PCR amplified DNA of the 16 S rRNA gene per sample, so if a species is only represented by on the order of 10 of those molecules (0.1%), then a single sequencing read error would be a 10% error, whereas a species represented by (>100) reads (1%) would only have 1% error per read.

### Choice of sample sizes

Sample sizes were chosen based on limitations of experimental throughput as increased number of biological replicates would have reduced the number of possible different communities that could be observed. We chose a minimum of two biological replicates (for complex communities in our validation set) and some sample types have up to seven biological replicates (such as the full community, which was repeated in most experiments as a control for consistency between experimental days).

## Computational methods

### Long short-term memory for dynamic prediction on microbial communities

Long short term memory (LSTM) networks belong to the class of recurrent neural networks (RNNs) and model time-series data. They were first introduced by Hochreiter et al. (*Hochreiter and Schmidhuber, 1997*) to overcome the vanishing or exploding gradients problem (*Hochreiter, 1998*) that occur due to long-term temporal dependencies. Since their inception, LSTMs have been further refined (*Gers et al., 2000*; *Graves and Schmidhuber, 2005*) and find numerous applications in several domains, including but not limited to neuroscience (*Storrs and Kriegeskorte, 2019*), weather forecasting (*Karevan and Suykens, 2020*), predictive finance (*Fischer and Krauss, 2018*), Google Voice for speech recognition (*Esch, 2014*; *Fischer and Krauss, 2018*) and Google Allo for message suggestion (*Wei et al., 2018*).

Similar to any recurrent neural network, an LSTM network comprises of a network of multiple LSTM units, each representing the input-output map at a time instant. *Figure 2* shows the schematic of the proposed LSTM network architecture for abundance prediction. For a microbial community comprising of $N$ species, each LSTM unit models the dynamics at time $t$ using the following set of equations:

$$i_t = \sigma\left(W_{ii}x_t + b_{ii} + W_{hi}h_{t-1} + b_{hi}\right)$$
$$f_t = \sigma\left(W_{if}x_t + b_{if} + W_{hf}h_{t-1} + b_{hf}\right)$$
$$g_t = \tanh\left(W_{ig}x_t + b_{ig} + W_{hg}h_{t-1} + b_{hg}\right)$$
$$o_t = \sigma\left(W_{io}x_t + b_{io} + W_{ho}h_{t-1} + b_{ho}\right) \quad (2)$$
$$c_t = f_t \odot c_{t-1} + i_t \odot g_t$$
$$h_t = o_t \odot \tanh(c_t),$$

where $h_t$, $c_t$, $x_t$ are the hidden state, cell state and input abundance at time $t$, respectively, and $i_t$, $f_t$, $g_t$, $o_t$ are input, forget, cell and output gates, respectively. $\sigma$ is the sigmoid function, and $\odot$ denotes the Hadamard product. The parameters $\{W_{mn}, b_{mn}\}$ for $m, n \in \{f, g, h, i, o\}$ are trainable and shared across all LSTM units. The output gate $o_t$ is further used to generate the abundance for next time instant as:

$$y_t : x_{t+1} = W_{yo}o_t + b_{yo}. \quad (3)$$

As shown in **Figure 2**, $y_t$ is fed to the LSTM unit at the next timestep ($t+1$), which in turn predicts the species abundance at time $t+2$. The process is repeated across multiple LSTM units in order to obtain $x_{t_{\text{final}}}$. The entire architecture is trained to minimize the mean-squared loss between the predicted abundance $x_{t_{\text{final}}}$ and true abundance $\hat{x}_{t_{\text{final}}}$.

## Using teacher forcing for intermittent time-series forecasting (*Figures 1, 2 and 5*)

The end-goal for the proposed LSTM-network based abundance predictor is to accurately capture the steady-state (final) abundance from initial abundance. In typical LSTM networks, the output of the recurrent unit at the previous timestep $y_{t-1}$ is used as an input to the recurrent unit at the current timestep $x_t$. This kind of recurrent model, while has the ability to predict final abundance, is incapable of handling the one-step-ahead prediction. The problem is even more critical when one tries to anticipate more than a single timestep into the future. *Teacher forcing* (**Benny Toomarian and Barhen, 1992**) entails a training procedure for recurrent networks, such as LSTMs, where 'true' abundances at intermittent timesteps are used to guide (like a teacher) the model to accurately anticipate one-step-ahead abundance.

Teacher forcing is an efficient method of training RNN models that use the ground truth from a prior time step as input. This is achieved by occasionally replacing the predicted abundance $y_{t-1}$ from the previous timestep with the true abundance $\hat{x}_t$ at the current timestep as input abundance to the LSTM unit at the current timestep during the training process. Teacher forcing not only stabilizes the training process, it *forces* the output abundances at all times to closely match the corresponding true abundances. This is precisely why we do not just use the ground truth abundances at intermittent timesteps in order to robustify the prediction of steady-state abundance. Once trained, the inference in such models is achieved by ignoring the ground truth abundances and using the predicted abundance from previous instant to roll forward the model in time. Teacher forcing was used to train the LSTM in all cases where intermediate time points were measured (**Figures 1, 2 and 5**), as opposed to cases that only included initial and final time points (**Figures 3 and 4**).

## Metabolite profiling

Microbial communities are a rich source of a variety of metabolites that are very commonly used as nutritional supplements, natural compounds to cure infectious diseases and in sustainable agriculture development. The concentration and chemical diversities of metabolites produced in a microbial community is a direct consequence of the diversity of interactions between organisms in the community. In essence, the dynamical evolution of relative species abundance and intra-community interactions govern the nature and amount of metabolites produced in the community. The functional map between species abundance and concentration of metabolites is highly complex and nonlinear, and is often approximated using simple regressors involving unary and pairwise interaction terms. In this paper, we model the species-metabolite map through appropriate modification of the LSTM network.

The aforementioned LSTM network for predicting the species abundance is suitably modified to augment four additional components that correspond to the concentration of metabolites at each

time instant. In particular, the species abundance data (of size $N_{\text{species}}$) is concatenated with the metabolite concentration data (of size $N_{\text{metabs}}$) to form a ($N_{\text{species}} + N_{\text{metabs}}$)-dimensional feature vector, which is suitably normalized so that the different components have zero mean and unity variance. The feature scaling is important to prevent over reliance on features with a broad range of values. Concatenation of species abundance data and the metabolite concentration data ensures that the future trajectory of metabolite concentrations evolves as a function of both the species abundance, as well as the metabolite concentrations at previous time instants. As before, the ($N_{\text{species}} + N_{\text{metabs}}$)-dimensional output of each LSTM unit is fed into the input block of the subsequent LSTM unit in order to advance the model forward in time. The model predictions at each time point is then transformed back to the original scale in order to obtain the Pearson $R^2$ scores on the unnormalized data. Compared with existing approaches that employ ordinary differential equations (ODEs) and multiple linear regression models for predicting metabolites, the proposed architecture enables more accurate and rapid estimation of all four metabolites. All the LSTM models were implemented in Python using PyTorch on an Intel i7-7700HQ CPU @2.80 GHz processor with 16 GB RAM and NVIDIA GeForce GTX 1060 (6GB GDDR5) GPU. The exact details of the neural network architecture consisting of number of layers, learning rate, choices of optimizer and nonlinear activations are described in *Supplementary file 1*.

## Data preprocessing for LSTM networks

Data normalization is one of the most widely adopted practices for efficient training of neural networks. Data normalization is known to speed up the training leading to faster convergence. At the same time, when working with multi-modal data or data with features represented at multiple scales, it is recommended to normalize the features (also known as feature standardization) to the same scale in order to avoid over reliance on features with large magnitudes. A common choice for feature standardization is to have zero-mean and unit-variance for each feature in the data. Let $x_i^{(n)}(k)$ and $c_j^{(n)}(k)$ represent the abundance of the $i^{\text{th}}$-species and concentration of the $j^{\text{th}}$-metabolite at the $k^{\text{th}}$ time instant for the $n^{\text{th}}$ sample. The mean and standard deviation of the quantities $\{x_i^{(n)}(k)\}$ and $\{c_j^{(n)}(k)\}$ can then be computed over the training dataset defined by:

$$\mu_{x_i}(k) = \frac{1}{N_{\text{samples}}} \sum_{n=1}^{N_{\text{samples}}} x_i^{(n)}(k), \qquad \mu_{c_j}(k) = \frac{1}{N_{\text{samples}}} \sum_{n=1}^{N_{\text{samples}}} c_j^{(n)}(k),$$

$$\sigma_{x_i}(k) = \sqrt{\frac{1}{N_{\text{samples}}} \sum_{n=1}^{N_{\text{samples}}} (x_i^{(n)}(k) - \mu_{x_i}(k))^2}, \qquad \sigma_{c_j}(k) = \sqrt{\frac{1}{N_{\text{samples}}} \sum_{n=1}^{N_{\text{samples}}} (c_j^{(n)}(k) - \mu_{c_j}(k))^2}.$$

The quantities can then be standardized as:

$$\tilde{x}_i^{(n)}(k) \colon \frac{x_i^{(n)}(k) - \mu_{x_i}(k)}{\sigma_{x_i}(k)},$$

$$\tilde{c}_j^{(n)}(k) \colon \frac{c_j^{(n)}(k) - \mu_{c_j}(k)}{\sigma_{c_j}(k)}.$$

The process is repeated for all species and metabolites at each time-point, and the scaled inputs $\{\tilde{x}_i^{(n)}(k)\}$ and $\{\tilde{c}_j^{(n)}(k)\}$ are then fed to the LSTM neural networks for prediction. During inference on the test data, the normalized output of each LSTM unit is inversely transformed back to its original scale using the precomputed $\{\mu_{x_i}(k), \sigma_{x_i}(k)\}$ and $\{\mu_{c_j}(k), \sigma_{c_j}(k)\}$. The readers are encouraged to refer to *Goodfellow et al., 2016*; *Zheng and Casari, 2018* for additional details on feature standardization. It is also not uncommon to normalize the data with respect to the mean and standard deviation of the entire dataset (and not just with respect to the training dataset). However, in practical scenarios, the test data is not known a priori, and thus it is undesirable to employ statistics from the hold-out test data for feature standardization. In our implementation of data preprocessing for LSTM networks, we had employed feature standardization using (a) training data only, and (b) both training and test data. The predictive performance of our LSTM models was nearly identical for both of these feature standardization approaches. The feature standardization can be toggled by the normalize_all variable in our open-source implementation.

## Hyperparameter tuning for LSTM networks

Similar to other learning algorithms, training an LSTM network entails choosing a set of hyperparameters for optimal performance. We used an exhaustive grid-search for hyperparameter optimization, while the choice of learning algorithm (optimizer) was restricted to Adam (*Kingma and Ba, 2014*) due to its superior empirical performance. For each experiment, nearly 10% of the training dataset was reserved as a cross-validation set, and the performances of the trained models were evaluated using cross-validation sets. This information was used to select the best hyperparameter settings. The choices for hyperparameters include: (a) learning rate, (b) number of hidden layers per LSTM unit, (c) number of units per layer within an LSTM unit, (d) mini-batch size, and (e) input data normalization. The input features are normalized to have zero mean and unit variance. The choices of learning rates include 0.005, 0.001, and 0.0001, respectively, each with a decay of 0.25 after every 25 epochs. The gradual decay in learning rates prevents potential overfitting to the data. An L2 regularization term with a very small weight decay coefficient ($10^{-5}$) is augmented to the loss function for preventing further overfitting. Choices for mini-batch size included 1, 10, 20, and 50, respectively. It was observed that sizes 10 and 20 resulted in improved training loss, and hence we used mini-batch sizes of 10 or 20 in all evaluations. The number of hidden layers per LSTM cell was iterated from 1 to 2. A two-layered LSTM did not result in any noticeable improvement over a single-layered LSTM cell, and thus we restricted our focus to just a single-layered LSTM cell for the sake of simplicity and faster training/inference. Finally, we tried 512, 1024, 2048, and 4096 hidden units per LSTM cell, and depending upon the complexity of the problem, we used 2048 hidden units (predictions of species and no prediction of metabolites) or 4096 hidden units (simultaneous prediction of species and metabolites). The exact details on the number of training epochs, learning rates, decay rates for different experiments can be found in the *Supplementary file 1*.

## Specific applications of computational methods

### Comparison of gLV and LSTM in silico (*Figure 1*)

To compare the LSTM and gLV models, we used a ground truth model of a 25-species community of the form:

$$\frac{dx_i(t)}{dt} = \left[ r_i + \sum_{\substack{j=1 \\ j \neq i}}^{N_{species}} a_{ij}x_j(t) + \sum_{\substack{j=1 \\ j \neq i}}^{N_{species}} \sum_{\substack{k=1 \\ k \neq i,j}}^{N_{species}} b_{ijk}x_j(t)x_k(t) \right] x_i(t), \tag{4}$$

where and represent individual species exponential growth rate and pairwise interaction coefficients, respectively. The parameters represent the effect of third-order interactions. The parameters and were derived from a gLV model in a previous study (*Clark et al., 2021*). We consider three types of simulation studies, each corresponding to varying contributions of the third-order interactions (second-order only: , mild third-order: uniformly sampling in the range , moderate third order: uniformly sampling in the range ). In each scenario, the ground truth data was generated for 624 training communities (25 monospecies, 300 two-member, 100 three-member, 100 five-member, and 99 six-member communities) by simulating the species abundance trajectories over the course of 48 hr and 'sampling' every 8 hr. The values from these 'sampled' time points were used to train both an LSTM model (methods described above in 'Computational Methods' with specific details in *Supplementary file 1*) and a standard gLV model (trained using FMINCON function in MATLAB as described previously [*Clark et al., 2021*]). These two models were then used to predict a set of 3299 hold-out communities >10 species simulated using the same ground truth model. For an additional analyses, the training data for the LSTM was augmented by including ground truth data for an additional 100 communities with 11 or 19 species.

### LSTM training for experimental 12-species community (*Figure 2*)

The data used in this analysis consisted of 175 microbial community subsets of a 12-species community sampled every 12 hr for 60 hr (*Venturelli et al., 2018*). Of these communities, 102 were chosen randomly to constitute the training data and the remaining 73 constituted the hold-out set. The LSTM was trained as described above in 'Computational Methods' with specific details in *Supplementary file 1*.

## Using LSTM Model to design multifunctional communities (*Figure 3*)

We used the LSTM model trained on previous data (*Figure 3a*) to design two sets of communities: a 'distributed' community set and a 'corner' community set. For the 'distributed' community set, we first took the predicted metabolite concentrations for all communities with .10 species and used $k$-means clustering with $k = 100$ (Python 3, scikit-learn v0.23.1, sklearn.cluster.Kmeans function) to identify 100 cluster centroids that were distributed across all of the predictions. We then found the closest community to each centroid in terms of Euclidean distance in the four-dimensional metabolite concentration space. These 100 communities constituted the 'distributed' community set.

For the 'corner' community set, we first defined four 'corners' in the lactate and butyrate concentration space by binning all communities with .10 species as shown in *Figure 3b*:

1. 5% lowest lactate concentration communities, then 5% lowest butyrate concentration of those
2. 5% lowest lactate concentration communities, then 5% highest butyrate concentration of those
3. 5% lowest butyrate concentration communities, then 5% lowest lactate concentration of those
4. 5% lowest butyrate concentration communities, then 5% highest lactate concentration of those.

Within each of those four 'corners', we identified four 'sub-corners' in the acetate and succinate concentration space by binning communities as shown in *Figure 3b*:

1. 5% lowest acetate concentration communities, then 5% lowest succinate concentration of those
2. 5% lowest acetate concentration communities, then 5% highest succinate concentration of those
3. 5% lowest succinate concentration communities, then 5% lowest acetate concentration of those
4. 5% lowest succinate concentration communities, then 5% highest acetate concentration of those

This process resulted in 16 'sub-corners' total. For each 'sub-corner', we then chose a random community and then identified four more communities that were maximally different from that community in terms of which species were present (Hamming distance). This overall process resulted in 80 communities constituting the 'corner' community set.

## Composite model: gLV model for predicting species abundance (*Figure 3*)

To benchmark the performance of the LSTM model for predicting metabolite production, we used a previously described Composite Model consisting of a generalized Lotka-Volterra (gLV) model for predicting species abundance dynamics and a regression model with interaction terms to predict metabolite concentration at a given time from the species abundances at that time (*Clark et al., 2021*). Because our LSTM model was trained on the same dataset as Composite Model M3 from *Clark et al., 2021*, we used those gLV model parameters.

## Composite model: regression models for predicting metabolite concentrations (*Figure 3*)

Our composite model implementation is similar to the model described in *Clark et al., 2021* for predicting metabolite concentration from community composition at a particular time. We used the exact gLV model parameter distributions obtained by *Clark et al., 2021*, which were determined using an approach based on *Shin et al., 2019*. The regression model mapping endpoint species abundance to metabolite concentrations from *Clark et al., 2021* was focused specifically on the prediction of butyrate. Therefore, we adapted the approach to prediction of multiple metabolites. First, we modified the model form to include first order and interaction terms for all 25 species, rather than just the butyrate producers. Then, we separately trained four regression models, one for each metabolite (butyrate, lactate, acetate, succinate), using the measured species abundance and measured metabolite concentrations from the same dataset used to train the LSTM model. We trained these models as described previously (*Clark et al., 2021*) by using Python scikit-learn (*Pedregosa, 2011*) and performed L1 regularization to minimize the number of nonzero parameters. Regularization coefficients were chosen by using 10-fold cross validation. We selected the regularization coefficient value with the lowest median mean-squared error across the training splits.

For predicting end-point metabolite profiles from initial species abundance using the LSTM network, a feed-forward network (FFN) was used at the output of the last LSTM unit to convert end-point species abundance to end-point metabolite concentrations. On the other hand, the composite model

in *Clark et al., 2021* uses multiple linear regressors at its output to predict a given metabolite concentration from species abundance. At this point, it is still unclear if the superior performance of LSTM network is due to the addition of a more powerful FFN at its output over the multiple linear regressors at the output of the composite model. Therefore, we replaced the simple multiple linear regressors component of the composite model from *Clark et al., 2021* with a Random Forest regressor (*Segal, 2004*) or a FFN. However, neither of these additions improved the metabolite prediction accuracy beyond that of the LSTM with a FFN (*Figure 3—figure supplement 3a*).

## Composite model: simulations for prediction (*Figure 3*)

Custom MATLAB scripts were used to predict community assembly using the gLV model as described previously (*Clark et al., 2021*). For each community, the growth dynamics were simulated using each parameter set from the posterior distribution of the gLV model parameters. The resulting community compositions for each simulation at 48 hr were used as an input to the regression models (multiple linear regression/Random Forest/FFN) implemented in Python to predict the concentration of each metabolite in each community for each gLV parameter set. Because of the large number of communities and the large number of parameter sets (i.e., hundreds of simulations per community), we used parallel computing (MATLAB parfor) to complete the simulations in a reasonable timeframe (~1 hr for the communities in *Figure 3—figure supplement 3a*).

## Understanding relationships between variables using LIME (*Figure 3*)

Black-box methods, such as the LSTM-networks employed in this manuscript, do not offer much insights into the underlying mechanics that make them so powerful. Consequently, any potential pitfalls that may come along with building such models remain unexplored. For networks that are of significant biological importance, basing assumptions on falsehoods can be catastrophic. We overcome this limitation by resorting to Local Interpretable Model-Agnostic Explanations (LIME) (*Ribeiro et al., 2016a*).

LIME has three key components: (a) *Local*, that is, any explanation reflects the behavior of a classifier around the sampled instance, (b) *Interpretability*, that is, the explanations offered by LIME are interpretable by human, (c) *Model-Agnostic*, that is, LIME does not require to peak into any model. It generates explanations by analyzing the model's behavior for an input perturbed around its neighborhood. In this manuscript, we employ LIME to explain both qualitatively and quantitatively, as to how the abundances of various species affect the concentrations of all four metabolites, and if the presence or absence of a given species has any significance on the resulting metabolite profile.

We carried out the LIME analysis to generate interpretable prediction explanations for model M2 for each community instance used to train the model. We used lime v0.2.0.1 for Python 3 (https://github.com/marcotcr/lime; *Ribeiro, 2021*) to train an explainer on the predictions of the training instances for each output variable (25 species, 4 metabolites) and then generated explanation tables for every input variable (species presence/absence) for every training instance. We then determined the median value for which the presence of a given species explained the prediction for each output variable to generate the networks in *Figure 3d, e*.

In a separate analysis, we investigated the sensitivity of LIME explanations to the training data used to fit the LSTM model. Because the purpose of this analysis was to understand the variability in LIME explanations and not to understand the dependence of LIME explanations on different communities, we only considered LIME explanations of the full community. This is in contrast to the LIME explanations shown in *Figure 3d, e*, which present the median LIME explanation taken over all of the communities in the training data. Training data was varied using 20-fold cross-validation, and LIME sensitivity of both metabolites (*Figure 3—figure supplement 5*) and species (*Figure 3—figure supplement 6*) was computed after fitting the LSTM model to each partition of the training data.

## Understanding relationships between variables using prediction sensitivity (*Figure 4*)

For each metabolite (Acetate, Butyrate, Lactate, Succinate), fractions of 0.5, 0.6, 0.7, 0.8, 0.9, and 1 of the total dataset were randomly sampled. Each sub-sampled dataset was subject to 20-fold cross validation to determine the sensitivity of held-out prediction performance to the amount of data available

for training. This process was repeated 30 times, and the average prediction over the 30 trials was used to compute the final held-out prediction performance ($R^2$).

The sensitivity of the model to the presence of individual species and pairs of species was determined by evaluating prediction performance ($R^2$) for subsets of the data containing each species and each possible pair of species. To evaluate how prediction performance of each metabolite was affected by the presence of species pairs, we computed the average percent difference between prediction performance taken over subsets containing a single species and all pairs of species using the following equation:

$$\text{Pairwise sensitivity} = \frac{100}{N_{species}^2} \sum_{i=1}^{N_{species}} \sum_{j \neq i}^{N_{species}} \frac{R_{ij}^2 - R_i^2}{R_i^2}, \tag{5}$$

where $R_i^2$ is the prediction performance taken over the subset of samples containing species , and $R_{ij}^2$ is the prediction performance taken over the subset of samples containing species  and $j$.

## Clustering metabolite trajectories (*Figure 5*)

To generate the clusters from the dynamic community observations (*Figure 5*), we used a graph-theoretic divisive clustering algorithm (*Jain et al., 1999*) based on the minimal spanning tree (*Zahn, 1971*). We first generated an undirected graph wherein each node was a community observed in our experiment and each edge weight was the Euclidean distance between two communities based on all metabolite measurements (4 metabolites ×3 time points = 12-dimensional space for Euclidean distance calculation). We then determined the minimal spanning tree for this graph using the minimum_spanning_tree function in networkx (v2.1) for Python 3. We then used this minimal spanning tree to generate clusters by iteratively removing the edge with the largest weight until 6 clusters were formed. In each iteration, if any edge removal resulted in a cluster with <5 communities (i.e. minimum cluster size), that edge was returned and the next largest edge was removed. The number of clusters and minimum cluster size were chosen based on an elbow method (*Pal and Biswas, 1997*), wherein scatter plots were made of the mean intracluster distance versus the number of clusters for various minimum cluster sizes and a combination of minimum cluster size and number of clusters that fell on the elbow of the plot was chosen.

## Decision tree classification of metabolite trajectories (*Figure 5*)

The decision tree shown in *Figure 5—figure supplement 1d* and used to produce the annotations in *Figure 5a* was generated using the DecisionTreeClassifier with the default parameter settings in scikit-learn (v0.23.1) for Python 3 (visualization generated using plot_tree function from the same).

## Understanding relationships between variables using sensitivity gradients (*Figure 5*)

Interpretability of neural-network (NN) models continues to be an interesting challenge in machine learning. While LIME is a great tool to explain what machine learning classifiers are doing, it is model-agnostic and uses simple linear models to approximate local behavior. Model-agnostic characteristic enforces retraining linear models on the training data and analyzing local perturbations, before LIME can be used to invoke interpretability. Moreover, the type of modifications that need to be performed on the data to get proper explanations are typically use case specific. Consequently, model-aware interpretability methods that take into account the weights of an already trained NN are more suitable.

For tasks, such as classification of images and videos, there is a natural way to interpret NN models using class activation maps (CAMs) (*Selvaraju et al., 2017*). CAMs assign appropriate weighting to different convolutional filters and highlights part of the images that activate a given output class the most. However, CAMs do not extend to other NN architectures, such as LSTMs. Fortunately for us, the answer to interpretability lies in the model training itself. Let $Y$ be the output variable of interest whose perturbation with respect to an input $x$ needs to be estimated. The effect of $x$ on $Y$ can be approximated through the partial derivative $\frac{\partial Y}{\partial x}$. For instance, $Y$ may denote butyrate concentration in an experiment, while $x$ can be used to represent abundance of a given species. The sign of the partial derivative depicts positive (or negative) correlation between the two variables, while the magnitude represents the extent of it. In order to evaluate the partial derivatives, we freeze the weights of the

already trained LSTM model and declare the inputs to be variables. A single backpropagation pass then evaluates the partial derivatives of an output variable of interest with respect to all the input variables. This is in contrast to LIME-based interpretability method, which requires training an additional model on top of an already trained deep learning model. Most deep learning libraries already implement a computational graph for performing efficient forward and backward passes during the training phase. This computational graph can be used to evaluate sensitivity gradients.

There indeed are other methods for explanation of neural networks, most notably the Shapley explainability method (*Lundberg and Lee, 2017*). This method is substantially more computationally burdensome than LIME or a sensitivity gradient based method (*Jia, 2019a*; *Jia, 2019b*). LIME and sensitivity gradients are based on first-order perturbations around the already learned model, and can be used to depict local model behavior with little to no computational burden. By contrast, explainability methods like Shapley are computationally expensive. An exact computation of Shapley values for a $K$-dimensional input requires estimating $2^K$ possible coalitions of the feature values and the "absence" of a feature has to be simulated by drawing random instances, which increases the variance for the estimate of the Shapley values estimation.

## Comparison of the discretized gLV model to the LSTM (*Figure 5—figure supplement 3*)

To train the gLV model using the same algorithm used to train the LSTM and enable metabolite prediction, the gLV model was discretized and augmented with a feed-forward neural network. The approximate gLV model is

$$x_i^{(t+1)} = x_i^{(t)} \left( r_i + \sum_{j=1}^{N_{species}} a_{ij} x_j^{(t)} \right),$$
$$c_i^{(t+1)} = \text{FFN}(x_i^{(t+1)}),$$

where $x_i$ is the abundance of species , $r_i$ is the growth rate of species , $a_{ij}$ represents the impact of species $j$ on species , and $c_i$ is the concentration of metabolite . Metabolite concentrations at time step $t$ are predicted using a feed-forward neural network (FFN). The structure of the discretized gLV model requires that all species abundances are strictly non-negative. When training the gLV, the data were pre-processed such that each feature (species abundance and metabolite concentration) ranges between zero and one based on the maximum value of each feature in the training data, computed at each time step. This is in contrast to the scaling used for the LSTM, which results in negative values for transformed species abundances. The stochastic gradient descent algorithm was used to train the LSTM and the discretized gLV model, using the Adam (*Kingma and Ba, 2014*) optimizer. The default settings of the Pytorch function, ReduceLROnPlateau, were used to adjust the learning rate during training. Species growth rates, $r_i$, and interaction coefficients, $a_{ij}$, were initialized to zero prior to fitting.

## Acknowledgements

This research was supported by funding from the Army Research Office (ARO) under grant number W911NF1910269, the National Institutes of Health under grant numbers R35GM124774 and R01EB030340. RLC was supported in part by an NHGRI training grant to the Genomic Sciences Training Program (T32 HG002760).

## Additional information

### Competing interests
Mayank Baranwal: The authors declare no conflicts of interest. Ryan L Clark: The other authors declare that no competing interests exist.

## Funding

| Funder | Grant reference number | Author |
|---|---|---|
| National Institutes of Health | R35GM124774 | Ophelia S Venturelli |
| Army Research Office | W911NF1910269 | Alfred O Hero<br>Ophelia S Venturelli |
| University of Wisconsin-Madison | R01 EB030340 | Ophelia S Venturelli |

The funders had no role in study design, data collection and interpretation, or the decision to submit the work for publication.

## Author contributions

Mayank Baranwal, Conceptualization, Data curation, Investigation, Methodology, Software, Validation, Visualization, Writing – original draft, Writing – review and editing; Ryan L Clark, Conceptualization, Funding acquisition, Investigation, Methodology, Software, Validation, Visualization, Writing – original draft, Writing – review and editing; Jaron Thompson, Data curation, Investigation, Methodology, Software, Validation, Visualization, Writing – original draft, Writing – review and editing; Zeyu Sun, Investigation, Software; Alfred O Hero, Conceptualization, Funding acquisition, Investigation, Methodology, Resources, Supervision, Visualization, Writing – original draft, Writing – review and editing; Ophelia S Venturelli, Conceptualization, Data curation, Funding acquisition, Investigation, Methodology, Project administration, Resources, Software, Supervision, Validation, Visualization, Writing – original draft, Writing – review and editing

## Author ORCIDs

Mayank Baranwal (iD) http://orcid.org/0000-0001-9354-2826
Ryan L Clark (iD) http://orcid.org/0000-0001-7865-2496
Jaron Thompson (iD) http://orcid.org/0000-0001-5967-0234
Alfred O Hero (iD) http://orcid.org/0000-0002-2531-9670
Ophelia S Venturelli (iD) http://orcid.org/0000-0003-2200-1963

## Decision letter and Author response

Decision letter https://doi.org/10.7554/eLife.73870.sa1
Author response https://doi.org/10.7554/eLife.73870.sa2

# Additional files

## Supplementary files

• Transparent reporting form

• Supplementary file 1. Modeling information. Contains two tables. Table S1 summarizes the training/test data and hyperparameters for training each LSTM model. Table S2 contains parameters used for all statistical tests presented in this work.

## Data availability

Pytorch implementation of the proposed LSTM model and the accompanying measurements of community composition and metabolite concentrations are available from GitLab (https://gitlab.eecs.umich.edu/mayank.baranwal/Microbiome; copy archived at swh:1:rev:f2eed8f013e42fbc-c8c8f98816a8146559d6f1b3). The raw Illumina sequencing data is available from Zenodo (https://zenodo.org/record/5529327, https://doi.org/10.5281/zenodo.5529327).

The following datasets were generated:

| Author(s) | Year | Dataset title | Dataset URL | Database and Identifier |
|---|---|---|---|---|
| Baranwal M | 2021 | Microbiome | https://gitlab.eecs.umich.edu/mayank.baranwal/Microbiome | GitLab, Microbiome |

*Continued on next page*

*Continued*

| Author(s) | Year | Dataset title | Dataset URL | Database and Identifier |
|-----------|------|---------------|-------------|-------------------------|
| Clark R | 2021 | NGS Data Accompanying "Deep Learning Enables Design of Multifunctional Synthetic Human Gut Microbiome Dynamics | https://doi.org/10.5281/zenodo.5529327 | Zenodo, 10.5281/zenodo.5529327 |

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
