## [Editor Report]

The ultimate goal of this work is to apply machine learning to learn from experimental data on temporal dynamics and functions of microbial communities to predict their future behavior and design new communities with desired functions. Using a significant amount of experimental data, the authors suggest a method that outperforms the state-of-the-art approach. The work is of broad interest to those working on microbiome prediction and design.

---

## [Decision Letter]

**Decision letter after peer review:**

Thank you for submitting your article "Deep Learning Enables Design of Multifunctional Synthetic Human Gut Microbiome Dynamics" for consideration by *eLife*. Your article has been reviewed by 3 peer reviewers, and the evaluation has been overseen by a Reviewing Editor and Aleksandra Walczak as the Senior Editor. The following individual involved in review of your submission has agreed to reveal their identity: Lingchong You (Reviewer #2).

Essential revisions:

1) All three reviewers agreed that not enough detail of the methods was provided. Please include this in the methods section, as well as an in-depth discussion and justification as to the choices made, including why LIME and CAM have been chosen as opposed to other methods, what are the inputs and outputs to the system, how the training and testing sets were chosen and how parameters for the two models were chosen.

2) The reviewers also agreed that the comparison of the performance of gLV to LSTM was not quite fair, due to many differences between the two implementations. To more fairly compare gLV to the LSTM, please augment the PyTorch code to have LSTM+FF and gLV+FF.

3) Carefully consider the language used throughout the paper, for example in the use of the term "deep learning".

*Reviewer #1 (Recommendations for the authors):*

Results, first paragraph. 'The experimental102 data was split into non-overlapping training and hold-out test sets, and an appropriate LSTM network was trained to predict species abundances at various time points given the information of initial species abundance. The details on the train/test split and the number of model hyperparameters are provided in Table S2'.

It is worth mentioning these details in the main text, at least regarding data division and architecture of the LSTM based model.

What was the motivation behind the particular division of the dataset: 624 training communities and 3299 testing communities? Why 6 or fewer species were used in the training and > = 10 were used in the testing dataset?

The authors state that since the glv model was used to generate the simulated data, they were not surprised to see that it predicts very well on this dataset. Would it be possible to use another benchmark to compare the two models?

Regarding the computational time of the LSTM model, I imagine it takes a lot of time to tune the hyper-parameters of the LSTM model. I think it should be mentioned that the 2 minutes required to train the LSTM model does not include this computational overhead. It is not clear how hyper-parameter tuning (number of layers, nodes, learning rate, activation function, etc.) was performed. Could you please add further details about this in the method section?

It is not clear why a FFN was used at the output of the LSTM? I could not find any information in the methods section about this. Can you explain why this particular setting was chosen?

The following line is not clear: 'all the network weights are trained simultaneously'.

What was the motivation behind choosing CAM and LIME as the explanation method among all the existing explanation methods?

LIME is a heuristic based method. How did you verify the robustness of the explanations? Could you please comment on if you changed some parameters (such as number of variables) and how it affected the explanations? How can we be sure that the explanation provided by the linear LIME models is actually what the non-linear LSTM model is doing? Are we agreeing with the explanations just because it is consistent with the current knowledge?

*Reviewer #2 (Recommendations for the authors):*

My general impression is the work is very dense and key insights are somewhat masked due to the lack of clarity in places. At the highest level, I think I understood the major points the authors tried to deliver (see above). However, I feel the authors can improve the presentation of the basic methodology and the intuition on a few points, which I find myself just having to trust what the authors have to say. Providing such intuition could minimize the impression of some of the performance comparisons being ad hoc.

1. It is actually not entirely clear what exactly feeds into the LSTM (or the gLV) and what comes out during the training process. Figure 1 is supposed to explain that but I could only gather that they're predicting time series. More specifically, what are the input variables? How many time series are being predicted? The authors might want to revise/expand the schematic to better explain the basic methodology.

Also, an overlay of a few representative true time courses vs. predicted time courses (e.g. good predictions vs. failed predictions) would be illuminating.

2. Figure 2 shows the prediction of gLV-model-simulated data using the LSTM or the gLV model. Only a brief description was provided, from which, I had a hard time understanding how the data were generated or predicted. For instance, the authors stated that a 25-member gLV model was used to make predictions. Those with < = 6 species were used to train the LSTM or the gLV model. Does it mean that, in these communities, only < = 6 species persisted after some time? And, the trained LSTM or gLV models only predict < = 6 species? If so, the following sentence becomes confusing. This confusion is related to the confusion above.

Also, how are higher-order interactions modeled in the gLV simulator?

3. It feels like the results in Figure 2 should go first (before Figure 1). The rationale is simple – for the simulated data, you have complete confidence in the ground truth, which can allow you to test the performance of the method in the idealized scenario. Also, the analysis on this data can probably allow you to provide an intuition when and why LSTM outperforms the gLV model (especially when the data were generated by another gLV model).

4. Also in Figure 2, since the data generation or training were not super time consuming, I wonder if the LSTM or gLV predictions could be drastically improved if larger training data sets are used. I'm actually quite surprised by the performance of LSTM on the simulated data, considering the somewhat limited size of the training set (624). A recent work had to use a much larger training set to achieve a high performance of LSTM (Wang et al., Nature Comm 2019) – this related work should be cited.

5. The overall performance of LSTM is impressive. It does fail occasionally even on simulated data (which is true for all ML models). To this end, the authors should discuss/show how to evaluate the confidence in individual predictions – checking each one (even if the ground truth is known) would defeat the purpose of using the neural network. This point is relevant when the trained neural network is being used to make predictions and probe biological insights.

6. I like how they use a side product of training procedure (backpropagation) to provide a quantitative metric of system sensitivity. I feel they could explain this point more.

7. For me, part of the reason the paper feels dense is that there are actually two stories (though closely related). One is the prediction of community dynamics. The other is the prediction of metabolite profiles. They represent somewhat different challenges in terms of evaluating LSTM performance.

The paper could have been split into two. Combining the two might have caused some of the confusion raised above.

*Reviewer #3 (Recommendations for the authors):*

Major comment: To make the comparison fair I would suggest using your torch code base to also obtain point estimates for the gLV parameters as well. There is no reason to use a sampling-based result from the prior study which uses a different model for metabolite prediction as well. Furthermore, the composite code is not accessible without a Matlab license (and thus not really reproducible). It is also trained in a two-step fashion (if this is not the case then there was not enough detail in the text to understand how the composite model was trained), and thus will be less accurate because of this. It should not take 5 hours for the model to learn the gLV parameters (your own reference [56] states that you have this down to minutes). Your torch code can be easily modified for the dynamics \log x_{t+1} = \log (x_t) + (diagonal(r) + A x_t) \δ_t and then this also means that you could have the same FF architecture for metabolites. This would allow for a much fairer comparison, both models could be trained in an identical fashion with the same metabolite prediction architecture. Although regularization and hyperparameter tuning would be slightly different of course.

Title and throughout: A single LSTM unit with a single hidden state (which can be of large dimension) is not really a deep network. With this interpretation a linear dynamical systems model with a single hidden state is a deep model as well.

21: In the abstract it is stated that current ODE models fail to capture complex behavior. This is not true. You fit an Ordinary difference equation to your data when solving for GLV, RNN, or LSTM (these are all ordinary because time is the only differential or difference variable). I think you meant to characterize the difference between parametric models like linear or gLV, etc, that are more "rigid" vs dynamical systems models that convolve functions up to arbitrary complexity (RNN, etc). Both are certainly classes of "Ordinary" difference or differential equations.

147: A gLV model is linear in the parameters so it is trivial to solve for the coefficients of the model in seconds. The text should be changed here. Also why does it take 5 hours to learn the gLV model when you state in [56] that you have this down to minutes?

Figure 1 (a) is this the model or the teacher forcing training procedure depicted. Also there is a subscript i, suggesting that there is no microbe-microbe interaction information captured by the model. Is the data in figure 1 all forecasted from t=0 or is this 1 sample look ahead prediction performance, it was not clear from the text?

Figure 2 (a) also has i subscript, was this intentional.

Methods:

– The LSTM model is not described in enough detail (what are the dimensions of the objects, how many hidden states [and why], how did you tune the hyperparameters of the model and the gradient solver [what gradient method did you use and why]).

– How was the composite model trained, was it indeed in a two-step fashion whereby the gLV model was trained and then a separate inference procedure was run for the metabolite model. More detail is needed here to understand what exactly was done and how the models were trained.

Plots in general:

– Plots only really show how the most abundant taxa are performing, would be nice to see the performance on the lower abundance taxa (maybe take a log of the OD?).

References:

You seem to be missing some important references from the Gerber Lab, "MDSINE Microbial Dynamical Systems INference Engine for microbiome time-series analyses" https://doi.org/10.1186/s13059-016-0980-6 is one such example but there are likely more given that lab primarily focuses on microbiome dynamics.

[Editors’ note: further revisions were suggested prior to acceptance, as described below.]

Thank you for resubmitting your work entitled "Deep Learning Enables Design of Multifunctional Synthetic Human Gut Microbiome Dynamics" for further consideration by *eLife*. Your revised article has been evaluated by Aleksandra Walczak (Senior Editor) and a Reviewing Editor.

The manuscript has been improved but there are some remaining issues that need to be addressed, as outlined below:

Reviewer #3 still finds that there is a need for a direct "apples-to-apples" comparison of gLV and LSTM. In your response to the reviewer's initial request, you are arguing that such a direct comparison would result in a very low performance of gLV. If this is the case, it would simply strengthen your argument and support the use of LSTM even more. It also did not seem like much additional work to carry out this analysis. If this is done and added as a supplement with some explanation of why it's not in the main text, I believe this would be satisfactory to all reviewers. Other than that, the reviewers were very appreciative of the modifications and the current quality of the paper.*Reviewer #1 (Recommendations for the authors):*

Thank you. This is exciting work, and the authors have addressed all of my concerns thoroughly.

*Reviewer #2 (Recommendations for the authors):*

The authors have fully addressed my comments on the original submission. I support its publication in the journal.

*Reviewer #3 (Recommendations for the authors):*

One outstanding issue with the paper remains. There is no apples-to-apples comparison of the LSTM+NN to a gLV+NN. I now comment on the authors' responses to an original comment.

[Authors] "We did not estimate new gLV parameters for this work, but rather used the gLV parameters that were already determined in [6]."

Having a composite model trained on different data and comparing it to a jointly trained model on entirely different data is not a sufficient comparison.

[Authors] "While we agree that the reviewer's proposed method for determining the gLV parameters would be a more direct comparison, there are several reasons why we chose to use the gLV parameters estimated in our previous study: (1) Higher resolution time-series measurements are needed to determine the exponential growth rate in the gLV model. Thus, estimating gLV parameters without the time-series measurements of monocultures would result in a poorly predictive model due to high uncertainty in the exponential growth rate parameters of the gLV model. Therefore, the proposed method of inferring parameters of the gLV model would yield a poorly predictive model."

If this is the case then the authors should show that the gLV model is not predictive with this sparse data. This begs the question though. Why did one need the LSTM model if the data doesn't really capture complex dynamics? The abstract stated that the LSTM was necessary for this very reason [Abstract]"Current models based on ecological theory fail to capture complex community behaviors due to higher-order interactions, do not scale well with increasing complexity and in considering multiple functions. We develop and apply a long short-term memory (LSTM) framework to advance our understanding of community assembly and health-relevant metabolite production using a synthetic human gut community." The authors are simultaneously arguing that the dynamics are too complex for simple gLV dynamics while also stating that the new time series are too simple to reliably train a gLV model. Is this a contradiction?

"(2) The methods used to estimate the gLV model parameters in our previous work were informed by higher resolution time-series measurements of monocultures (individual species). Thus, the gLV model had additional information beyond that of the LSTM model, which was only informed by the initial point and endpoint (with the exception of Figure 5)."

This statement is confusing. The LSTM model was trained using teacher forcing with intermediate time points per the methods section?

"(3) The optimization algorithm MATLAB FMINCON is a widely used approach for estimating parameters of the gLV model (see [6], [8] and [10], for example) (4) The proposed method of log transforming the gLV model only works with higher resolution time-series data. We have sparse time sampling in our datasets."

If indeed the data is so sparse that a traditional gLV model would not learn then it would be easy for the authors to show this. In order to be published Figure 4 should compare gLV+NN and LSTM+NN trained on the same data in the same way and compare the model's performance directly.

---

## [Author Response]

Essential revisions:1) All three reviewers agreed that not enough detail of the methods was provided. Please include this in the methods section, as well as an in-depth discussion and justification as to the choices made, including why LIME and CAM have been chosen as opposed to other methods, what are the inputs and outputs to the system, how the training and testing sets were chosen and how parameters for the two models were chosen.

We have addressed this comment throughout our response. here is a summary of our edits to the manuscript to address this issue: As we have commented elsewhere in this review, we have done the following to clarify the details of our modeling:

A. We have reorganized our methods section to make it easier to find relevant details. We have created three sections: “Experimental Methods”, “Computational Methods”, and “Specific Applications of Computational Methods”. This final section has new subsections describing all analyses presented in the paper with references to the specific Figure(s) that use the described methods.

B. We have added details about the ground truth models and train/test split used for ourin silico comparison of the gLV and LSTM in predicting species abundance in the section labeled “Comparison of gLV and LSTM in silico (Figure 1)”

C. We have clarified the Methods section describing the composite model used for comparison with the LSTM for predicting species abundance and metabolite production. Methods Section “Composite Model: Regression Models for Predicting Metabolite Concentrations (Figure 3)”.

D. We have reordered Figures 1 and 2 for a more logical flow and have edited these sections to make it easier for the reader to understand relevant details.

E. We have added additional justification for the use of LIME and CAM in the relevant places in the manuscript. This includes a new sensitivity analysis of the LIME results to different test/train splits of our data (Figure 3—figure supplement 5, Figure 3—figure supplement 6).

2) The reviewers also agreed that the comparison of the performance of gLV to LSTM was not quite fair, due to many differences between the two implementations. To more fairly compare gLV to the LSTM, please augment the PyTorch code to have LSTM+FF and gLV+FF.

We have updated the comparisons in Figure 3—figure supplement 3a to include the prediction accuracy for gLV+FF and gLV+Random Forest Regressor. While some improvements in the prediction of Succinate, Lactate, and Acetate were observed relative to the original composite model, the LSTM outperformed all of these models for all four metabolites. We have added text to describe this result in the main text:

“Additionally, replacing the regression portion of the composite model with either a Random Forest Regressor or a Feed Forward Network did not improve the metabolite prediction accuracy beyond that of the LSTM (Figure 3—figure supplement 3a).”

We have also added descriptions of these new methods to the manuscript in the section

“Composite Model: Regression Models for Predicting Metabolite Concentration (Figure 3)”

3) Carefully consider the language used throughout the paper, for example in the use of the term "deep learning".

The use of the terminology “deep learning” in the title and throughout the paper is consistent with accepted terminology used by the machine learning community for LSTM’s. Note that LSTMs are featured prominently as a mainstay category of deep learning networks in the definitive book on deep learning by Goodfellow, et al. [1, Ch. 10]. Please refer to our response 8 to Reviewer#3 for more details.

We have suitably revised the manuscript to improve readability. In particular, text corresponding to Figures 1 and 2 in the original manuscript have been reorganized for better readability. We have also added the following paragraph in the Discussion section to appropriately emphasize the scope of our work and some of its limitations.

“The current implementation of the LSTM model lacks uncertainty quantification for individual predictions, which could be used to guide experimental design [38]. Recent progress in using Bayesian recurrent neural networks has led to emergence of Bayesian LSTMs [39, 40], which provides uncertainty quantification for each prediction in the form of posterior variance or posterior confidence interval. However, currently, the implementation and training of such Bayesian neural networks can be significantly more difficult than training the LSTM model developed here. In addition, we benchmarked the performance of the LSTM against the standard gLV model which has been demonstrated to provide accurate predictions of community assembly in communities up to 25 species [6, 8]. The gLV model has been modified mathematically to capture more complex system behaviors [41]. However, while comparing such modified gLV models to the LSTM could further elucidate their differences, implementation of these gLV models to represent the behaviors of microbiomes with a large number of interacting species poses major computational challenges.”

Reviewer #1 (Recommendations for the authors):Results, first paragraph. 'The experimental102 data was split into non-overlapping training and hold-out test sets, and an appropriate LSTM network was trained to predict species abundances at various time points given the information of initial species abundance. The details on the train/test split and the number of model hyperparameters are provided in Table S2'.It is worth mentioning these details in the main text, at least regarding data division and architecture of the LSTM based model.

Thank you for the suggestion. We have included the following details in the main text in the revised version in Section titled “LSTM accurately predicts experimental microbial community assembly”.

“Of the 175 microbial communities, 102 microbial communities were selected randomly to constitute the training set, while the remaining 73 microbial communities constituted the hold-out test set.”

“Each LSTM unit consists of a single hidden layer comprising of 2048 hidden units with ReLU activation. The details on hyperparameter tuning, learning rates, and choice of optimizer are provided in the Methods section.”

What was the motivation behind the particular division of the dataset: 624 training communities and 3299 testing communities? Why 6 or fewer species were used in the training and > = 10 were used in the testing dataset?

This is an excellent question. This was done to demonstrate precisely the fact that our model generalizes to higher-order interactions. We train the model on simpler communities (≤ 6 species), and yet show that the model is able to capture microbial community dynamics in high richness communities (≥ 10 species). In natural communities, the number of species ranges between tens to thousands. In addition, the availability of sufficiently time-resolved measurements of community dynamics can be potentially very sparse.

The breakup of 624 communities is as follows: all 25 one-member communities, all 300 two-member communities, 100 (out of 2300 unique) three-member communities, 100 (out of 53130 unique) five-member communities and 99 (out of 177100) six-member communities. As evident, the total number of possible communities, even when one restricts to analyzing only simpler communities, is large since the total number grows exponentially with the number of species in the community. A similar training/test partitioning was used in our previous papers [5, 6, 7]. It is very common in other fields of science to ask the question of whether the behavior of lower-order systems (i.e. parts) can predict higher-order systems (whole system). Therefore, we apply a similar approach to microbial communities (e.g. similar questions have been asked in chemistry and physics).

We have added a detailed description on the rationale behind the choice of training/test sets in the section titled “LSTM outperforms the generalized Lotka Volterra ecological model” in the revised manuscript. We do not quote the edits here for the sake of brevity.

The authors state that since the glv model was used to generate the simulated data, they were not surprised to see that it predicts very well on this dataset. Would it be possible to use another benchmark to compare the two models?

We thank the reviewer for this question. If there was a widely accepted mechanistic model (beyond gLV) that described microbial community dynamics, we would have considered using this benchmark model instead of gLV for the comparison of the gLV and LSTM model in Figure 1. For example, dynamic flux balance analysis of genome-scale metabolic models can be used to generate time-resolved changes in species abundance. However, there are notable limitations of these models: (1) it can be challenging to use these models to study high species richness communities due to computational limitations, (2) these models have strict assumptions (e.g. metabolic fluxes are solved to optimize the growth rate of individual species), (3) genome sequences have missing or incorrect functional annotations especially for non-model bacterial species such as human gut isolates, (4) there is large uncertainty about the transport reactions of metabolites mediating interspecies interactions and (5) these models only capture inter-species interactions that arise from production/degradation of metabolites in metabolism and thus neglect other types of microbe-microbe interactions (e.g. environmental pH mediated interactions). Alternatively, Consumer-Resource models capture species growth dynamics as a function of resources uptake/utilization. However, these models generally do not capture other mechanisms beyond resource competition (e.g. metabolites that are produced by constituent members of the community and used as substrates by other members or production of toxins that inhibit species growth). Mechanisms of microbial interactions beyond resource competition are key determinants of community dynamics. In addition, these models are very difficult to apply in practice due to the myriad of unknown mechanisms of interaction driving microbe-microbe interactions. Therefore, while the gLV model does not capture the metabolites mediating inter-species interactions, it is a simplified ecological model that represents both positive and negative interactions shaping community dynamics. Despite its limitations, the gLV model is widely used by the microbiome research community and can accurately predict community assembly for synthetic microbiomes containing up to 25 species and elucidate significant inter-species interactions that were independently validated experimentally [6, 8, 7]. Finally, models that have been developed to augment the gLV with additional terms can be very challenging to implement in practice with experimental data.

Regarding the computational time of the LSTM model, I imagine it takes a lot of time to tune the hyper-parameters of the LSTM model. I think it should be mentioned that the 2 minutes required to train the LSTM model does not include this computational overhead. It is not clear how hyper-parameter tuning (number of layers, nodes, learning rate, activation function, etc.) was performed. Could you please add further details about this in the method section?

Thank you for your comments. The hyperparameters were tuned using a simple grid search. The hyperparameter choices include: (a) learning rate and its decay scheme for the Adam optimizer, (b) number of hidden layers per LSTM unit, (c) Number of units per layer within an LSTM unit, (d) mini-batch size, and (e) input data normalization. For feature normalization, we used the standard mean-zero, unit-variance normalization. The choices of learning rates included 0.005, 0.001 and 0.0001, respectively, each with a decay of 0.25 after every 25 epochs. Choices for mini-batch size included 1, 10, 20 and 50, respectively. It was observed that sizes 10 and 20 resulted in improved training loss, and hence we used mini-batch sizes of 10 or 20 in all our evaluations. The number of hidden layers per LSTM cell was iterated from 1 to 2. A two-layered LSTM did not result in any noticeable improvement over a single-layered LSTM cell, and thus we restricted our focus to just a single-layered LSTM cell for the sake of simplicity and faster training/inference. Finally, we tried 512, 1024, 2048 and 4096 hidden units per LSTM cell, and depending upon the complexity of the problem, we decided to work with either 2048 hidden units (no prediction of metabolites) or 4096 hidden units (simultaneous prediction of metabolites).

It must also be emphasized that the model hyperparameters were optimized for performance on a held-out cross-validation set (formed using randomly chosen 10% of the samples from the training data). Since the time taken for each run of the training process is only a couple of minutes, the entire process of tuning hyperparamters does not result in any significant computational efforts.

We have added the relevant details in the revised manuscript in a new subsection titled “Hyperparameter tuning for LSTM Networks”.

We would also like to emphasize that training gLV models, too, requires hyperparameter tuning. Fitting an ODE-based parametric model entails solving nonlinear optimization problems, the solution of which is highly dependent on the choice of initial guess parameters, choice of optimization algorithms – (a) active-set methods, (b) interior-point methods, or (c) trust region-reflective methods; and a prior knowledge of suitable parameter range. Additionally, fitting of ODE-based models also frequently involves selection of regularization parameters to reduce overfitting of the model to the noise in the experimental data [8] [6].

It is not clear why a FFN was used at the output of the LSTM? I could not find any information in the methods section about this. Can you explain why this particular setting was chosen?

We thank the reviewer for this request for clarification. We did not use an FFN at the output of the LSTM for predicting predicting species abundances. However, we did use an FFN as the final layer of the LSTM to predict metabolite concentrations. In considering your comment, we noticed two inaccuracies in the schematic representation of our models. In Figure 3a, we neglected to include the FFN in the schematic, whereas an FFN was incorrectly included in Figure 5h in our original submission. We have now corrected these figures in the revised paper.

Recall that species abundance is Figure 3 is a 25-dimensional vector comprising of absolute abundances of each of the species. On the other hand, metabolite profile is a four-dimensional vector corresponding to metabolite concentration of each of the four metabolites. The FFN layer converts the 25-dimensional vector at the output of the last LSTM unit to a four dimensional vector, and is trained to predict the metabolite profile. Thus, only when the species absolute abundance measurements are available at sampled time-points, an FFN is used to map the species abundance output to corresponding metabolite profile. We have clarified this statement in the revised version in the subsection entitled “LSTM enables end-point design of multifunctional synthetic human gut microbiomes” “This model uses a feed-forward network (FFN) at the output of the final LSTM unit that maps the end-point species abundance (a 25-dimensional vector) to the concentrations of the four metabolites (Figure 3a).”

The following line is not clear: 'all the network weights are trained simultaneously'.

We apologize for any confusion. The statement is added to emphasize that the model was trained in an end-to-end manner. We have reworded the phrase in the revised version in Section titled “LSTM enables end-point design of multifunctional synthetic human gut microbiomes”.

“The entire neural network model comprising LSTM units and a feed-forward network is learned in an end-to-end manner during the training process, (i.e., all the network weights are trained simultaneously).”

What was the motivation behind choosing CAM and LIME as the explanation method among all the existing explanation methods?

Please refer to our detailed response to your public review summary. Additional details on the choice of explain ability methods can be found in the Section titled “Understanding Relationships Between Variables Using LIME” in the revised manuscript.

LIME is a heuristic based method. How did you verify the robustness of the explanations? Could you please comment on if you changed some parameters (such as number of variables) and how it affected the explanations? How can we be sure that the explanation provided by the linear LIME models is actually what the non-linear LSTM model is doing? Are we agreeing with the explanations just because it is consistent with the current knowledge?

We thank the reviewer for the excellent question. Our work encompasses three-fold verification of LIME explanations – (a) As suggested by the reviewer, the explanations are consistent with the current knowledge based on our previous study [6], (b) the explanations are consistent with those of sensitivity-gradient method (Figure 5), (c) To demonstrate that LIME explanations of the full community (25-member) were robust, we performed new model analyses by partitioning the training data by 20-fold and training an LSTM model on each partition. In almost every case, the inter-quartile range of the strongest LIME explanations remained either strictly positive or negative, indicating that while the magnitude of LIME explanations varied, the sign (representing the type of interaction) was consistent. This implies that the LIME explanations are robust to variations in the amount of training data (see new Figure 3—figure supplement 5 and Figure 3—figure supplement 6). We have added the following text:

“We explored the consistency of LIME explanations for the full 25-member community in response to random partitions of the training data to provide insights into the sensitivity of the LIME explanations given the training data (Figure 3—figure supplement 5, Figure 3figure supplement 6). These results demonstrated that the direction of the strongest LIME explanations of the full community were consistent in sign despite variations in magnitude.”

Reviewer #2 (Recommendations for the authors):My general impression is the work is very dense and key insights are somewhat masked due to the lack of clarity in places. At the highest level, I think I understood the major points the authors tried to deliver (see above). However, I feel the authors can improve the presentation of the basic methodology and the intuition on a few points, which I find myself just having to trust what the authors have to say. Providing such intuition could minimize the impression of some of the performance comparisons being ad hoc.

We apologize for any confusion. We have made the following modifications to the text to clarify the presentation. (1) We have inverted the order of the first two sections as suggested in your comment below. (2) We have made substantial edits to the first two sections (highlighted in the revised manuscript) to streamline the information presented and clarify details of our methods. (3) We have added two new section headings to better guide the reader to the key takeaways: for the LIME Analysis: “Using local interpretable model-agnostic explanations to decipher interactions” and for the LSTM trained on higher time-resolution data: “Using LSTM with higher time-resolution to interpret contextual interactions”

1. It is actually not entirely clear what exactly feeds into the LSTM (or the gLV) and what comes out during the training process. Figure 1 is supposed to explain that but I could only gather that they're predicting time series. More specifically, what are the input variables? How many time series are being predicted? The authors might want to revise/expand the schematic to better explain the basic methodology.

We apologize for any confusion. Figure 2 (Figure 1 in the original submission) represents a scatter plot of all test communities (consisting of a 12-member synthetic human gut community and subsets of this community) across all species at different time instants. The input to our LSTM network is the initial species absolute abundance for each community. The LSTM predicts the absolute abundances of different species in the community at various time points. For the ease of illustration and brevity, the predicted abundances of species across 73 test communities (not included in the training data) are plotted against their true abundances.

We agree with the reviewer that it is illuminating to show representative communities that varied in their prediction accuracy by the LSTM network using the time-series data shown in Figure 2. As such, we have added a new Supplementary Figure (Figure 2—figure supplement 1) shows the predictions of the LSTM network for three (out of 73) representative communities: (a) well-predicted, (b) prediction score in the median range, (c) poorly-predicted. The prediction *R*^2^-score for each community is the coefficient of determination between measured (true) and predicted abundances of all species in that community at all time instants. A histogram of *R*^2^-scores across 73 test communities is shown in Figure 2—figure supplement 1a. These data show that most communities are predicted with significantly higher *R*^2^-scores than the median score being ≈ 0*.*76.

Additionally, we have also included plots showing the temporal evolution of each species in the three representative communities – (i) well predicted (*R*^2^ = 0*.*91, Figure 2—figure supplement 1b), (ii) median score (*R*^2^ = 0*.*76, Figure 2—figure supplement 1c), (iii) poorly-predicted (*R*^2^ = 0*.*14, Figure 2—figure supplement 1d). The test dataset, comprising of 73 communities, had only 15% communities with ≥ 5 species (majority were 2-4 member communities). We found that higher richness communities were accurately predicted by our LSTM network.

Also, an overlay of a few representative true time courses vs. predicted time courses (e.g. good predictions vs. failed predictions) would be illuminating.2. Figure 2 shows the prediction of gLV-model-simulated data using the LSTM or the gLV model. Only a brief description was provided, from which, I had a hard time understanding how the data were generated or predicted. For instance, the authors stated that a 25-member gLV model was used to make predictions. Those with < = 6 species were used to train the LSTM or the gLV model. Does it mean that, in these communities, only < = 6 species persisted after some time? And, the trained LSTM or gLV models only predict < = 6 species? If so, the following sentence becomes confusing. This confusion is related to the confusion above.Also, how are higher-order interactions modeled in the gLV simulator?

We again apologize for the confusion due to our lack of detailed descriptions of the in silico methods. We have substantially modified the section “LSTM outperforms the generalized Lotka Volterra ecological model” and added additional information to the Methods section describing these in silico experiments “Comparison of gLV and LSTM in silico”. To clarify the specified point referring to “communities with ≤ 6 species”: This means only 6 species are present with non-zero abundance at the initial time point (time 0). Thus, all other species should have zero abundance at later time points (i.e. species whose abundance at t=0 is zero do not spontaneously appear at later time points).

3. It feels like the results in Figure 2 should go first (before Figure 1). The rationale is simple – for the simulated data, you have complete confidence in the ground truth, which can allow you to test the performance of the method in the idealized scenario. Also, the analysis on this data can probably allow you to provide an intuition when and why LSTM outperforms the gLV model (especially when the data were generated by another gLV model).

Thank you very much for your suggestion. We, too, had initially considered adding Figure 2 (and accompanying text) before Figure 1. Upon further consideration, we agree that this is the more logical order and have rearranged these two sections in the text.

4. Also in Figure 2, since the data generation or training were not super time consuming, I wonder if the LSTM or gLV predictions could be drastically improved if larger training data sets are used. I'm actually quite surprised by the performance of LSTM on the simulated data, considering the somewhat limited size of the training set (624). A recent work had to use a much larger training set to achieve a high performance of LSTM (Wang et al., Nature Comm 2019) – this related work should be cited.

We agree with the reviewer that the performance of any data-driven algorithm improves with both the quantity and quality of available data. The reason we *purposely* worked with limited training data was primarily due to the fact that in a practical (experimental) setting, data collection is expensive and thus, any model, capable of working with limited data will prove to be most useful. Another compelling reason for working with limited data is to elucidate that LSTMs generalize better, i.e., LSTMs can translate learning on lower richness communities to predict the behavior of higher richness communities. Finally, the quality of training data also plays a significant role in the learned predictive behavior of the model. Suppose, our experimental budget allows us to perform a fixed number of limited experiments. Certainly, if each of these experiments consist of communities that do not involve a particular species ‘X’, and the model is then asked to predict the behavior of communities involving ‘X’, the model is bound to fail. Therefore, training data that includes all species and displays a wide range in the number of species in each community (wide range of species richness) should constitute the ideal learning scenario. Further, the training data should display sufficient variation in the abundance of each species or concentration of each metabolite to be informative for the model (see Figure 4—figure supplement 1). We have cited the Want et al., Nat Commun 2019 paper in our Discussion section:

“Achieving a highly predictive LSTM model required substantially less training data than a previous study that approximated the behavior of mechanistic biological systems models with RNNs (Figure 2) [9]. While the performance of any data-driven algorithm improves with the quantity and quality of available data, we demonstrate that the LSTM can translate learning on lower-order communities to accurately predict the behavior of higher-order communities given a limited and informative training set that is experimentally feasible. For synthetic microbial communities, the quality of the training set depends on the frequency of timeseries measurements within periods in which the system displays rich dynamic behaviors (i.e. excitation of the dynamic modes of the system), the range of initial species richness, representation of each community member in the training data and sufficient variation in species abundance or metabolite concentration (Figure 4—figure supplement 1).”

5. The overall performance of LSTM is impressive. It does fail occasionally even on simulated data (which is true for all ML models). To this end, the authors should discuss/show how to evaluate the confidence in individual predictions – checking each one (even if the ground truth is known) would defeat the purpose of using the neural network. This point is relevant when the trained neural network is being used to make predictions and probe biological insights.

Quantification of uncertainty in predictions is an excellent suggestion. However, like most other discriminative machine learning models, the LSTM produces predictions but is not equipped to quantify the uncertainty associated with a given prediction. We have endeavored in this paper to quantify the overall uncertainty over all training data using cross-validation analysis for assessing sensitivity of the LSTM predictions (Figure 4a) and of the LIME explanations for a representative community (25-member community, see new Figure 3—figure supplement 5 and Figure 3—figure supplement 6). However, to obtain a sample-by-sample prediction of uncertainty of the prediction requires additional modeling of uncertainty, e.g., provided by a Bayesian formulation of the LSTM, would be required, which is beyond the scope of this paper and an ongoing project in our lab. However, since we agree with the reviewer on the importance of quantification confidence, in the revision we have included the following comment near the end of the Discussion section:

“The current implementation of the LSTM model lacks uncertainty quantification for individual predictions, which could be used to guide experimental design. Recent progress in using Bayesian recurrent neural networks has led to emergence of Bayesian LSTMs, which provides uncertainty quantification for each prediction in the form of posterior variance or posterior confidence interval. However, currently, the implementation and training of such Bayesian neural networks can be significantly more difficult than training the LSTM model developed here. In addition, we benchmarked the performance of the LSTM against the standard gLV model which has been demonstrated to provide accurate predictions of community assembly in communities up to 25 species. The gLV model has been modified mathematically to capture more complex system behaviors. However, while comparing such modified gLV models to the LSTM could further elucidate their differences, implementation of these gLV models to represent the behaviors of microbiomes with a large number of interacting species poses major computational challenges.”

6. I like how they use a side product of training procedure (backpropagation) to provide a quantitative metric of system sensitivity. I feel they could explain this point more.

Thank you for your observation. Unlike LIME-based interpretability methods that require training an additional model on top of already trained deep-learning models, the proposed sensitivity analysis requires a single backpropagation pass to capture input-output behavior. The proposed method is thus computationally inexpensive, and produces similar explanations as LIME’s. We have emphasized this further in our revised version in the subsection entitled “Understanding Relationships Between Variables Using Sensitivity Gradients.”

7. For me, part of the reason the paper feels dense is that there are actually two stories (though closely related). One is the prediction of community dynamics. The other is the prediction of metabolite profiles. They represent somewhat different challenges in terms of evaluating LSTM performance.The paper could have been split into two. Combining the two might have caused some of the confusion raised above.

While we agree that combining the prediction of species abundance with metabolite concentrations makes the story more dense, we also would like to emphasize that these two problems are intimately connected. From a biological perspective, microbes continuously impact metabolites by reacting to these metabolites and releasing or consuming them. For example, metabolites impact microbial growth by being energy sources/building blocks, causing toxicity, etc. By modeling both microbial growth and the metabolites they produce/consume together, we can capture these key interconnections. This is particularly evident in the examples shown in Figure 5 where we see non-monotonic metabolite trajectories. These trends in metabolite concentrations are likely due to different phases of production/consumption of metabolites which could be captured in our model as feedback between the microbes and metabolite variables. We have revised our Discussion to mention this key aspect of our study.

“Microbial communities continuously impact metabolites by releasing or consuming them. Therefore, by modeling both microbial growth and the metabolites they produce/consume together, we were able to capture the interconnections between these variables.”

Reviewer #3 (Recommendations for the authors):Major comment: To make the comparison fair I would suggest using your torch code base to also obtain point estimates for the gLV parameters as well. There is no reason to use a sampling-based result from the prior study which uses a different model for metabolite prediction as well. Furthermore, the composite code is not accessible without a Matlab license (and thus not really reproducible). It is also trained in a two-step fashion (if this is not the case then there was not enough detail in the text to understand how the composite model was trained), and thus will be less accurate because of this. It should not take 5 hours for the model to learn the gLV parameters (your own reference [56] states that you have this down to minutes). Your torch code can be easily modified for the dynamics \log x_{t+1} = \log (x_t) + (diagonal(r) + A x_t) \δ_t and then this also means that you could have the same FF architecture for metabolites. This would allow for a much fairer comparison, both models could be trained in an identical fashion with the same metabolite prediction architecture. Although regularization and hyperparameter tuning would be slightly different of course.

We thank the reviewer for this comment. While we appreciate the reviewer’s desire for a more direct comparison, we chose to compare our LSTM framework to the current best implementation of the gLV modeling framework in our previous study [6]. Part of the confusion here may have been due to lack of clarity in our Methods section. We did not estimate new gLV parameters for this work, but rather used the gLV parameters that were already determined in [6].

While we agree that the reviewer’s proposed method for determining the gLV parameters would be a more direct comparison, there are several reasons why we chose to use the gLV parameters estimated in our previous study: (1) Higher resolution time series measurements are needed to determine the exponential growth rate in the gLV model. Thus, estimating gLV parameters without the time-series measurements of monocultures would result in a poorly predictive model due to high uncertainty in the exponential growth rate parameters of the gLV model. Therefore, the proposed method of inferring parameters of the gLV model would yield a poorly predictive model. (2) The methods used to estimate the gLV model parameters in our previous work were informed by higher resolution time-series measurements of monocultures (individual species). Thus, the gLV model had additional information beyond that of the LSTM model, which was only informed by the initial point and end point (with the exception of Figure 5). (3) The optimization algorithm MATLAB FMINCON is a widely used approach for estimating parameters of the gLV model (see [6], [8] and [10], for example) (4) The proposed method of log transforming the gLV model only works with higher resolution time-series data. We have sparse time sampling in our datasets.

We have modified the methods section referring the Composite Model to clarify this point:

“Our composite model implementation is similar to the model described in [8] for predicting metabolite concentration from community composition at a particular time. We used the exact gLV model parameter distributions from [8] which were determined using an approach based on [62].”

On the topic of the MATLAB license and reproducible results, we found in our previous work that we needed the tools provided in MATLAB for the task at hand. If an interested party without access to a MATLAB license wanted to reproduce these results, they could do so using GNU Octave (https://www.gnu.org/software/octave/index), a free software compatible with most MATLAB functions.

While [11] describes a method for estimating gLV parameters that takes only minutes, this method used only individual species and pairwise community growth data. In our work described in [6], we found that including communities with a higher number of species reduced the sparsity of the matrix computations and thus complicated the use of this method, so we adapted to use the MATLAB FMINCON function. This approach is fully described in [6] and does require hours to for parameter estimation.

Title and throughout: A single LSTM unit with a single hidden state (which can be of large dimension) is not really a deep network. With this interpretation a linear dynamical systems model with a single hidden state is a deep model as well.

While it is true that a single LSTM unit has a single hidden layer, it is common to implement a sequence of LSTM’s arranged in multiple layers for modeling time series data, a process known as unrolling, which produces a deep network. As explained in the text, and exhibited in Figures 1a and 2a, we are using such a multilayer LSTM, where each layer corresponds to a different time point of our data. Thus our use of the term “deep network” for our LSTM model is consistent with accepted terminology used by the machine learning community for LSTM’s. Note that LSTMs are featured prominently as a mainstay category of deep learning networks in the definitive book on deep learning by Goodfellow, et al. [1, Ch. 10].

21: In the abstract it is stated that current ODE models fail to capture complex behavior. This is not true. You fit an Ordinary difference equation to your data when solving for GLV, RNN, or LSTM (these are all ordinary because time is the only differential or difference variable). I think you meant to characterize the difference between parametric models like linear or gLV, etc, that are more "rigid" vs dynamical systems models that convolve functions up to arbitrary complexity (RNN, etc). Both are certainly classes of "Ordinary" difference or differential equations.

We apologize for the confusion here. The reviewer is correct in emphasizing that the point we were actually trying to make was the difference between parametric models based on ecological theory (like gLV) and data driven models such as the RNN and LSTM. We have modified the abstract to better reflect this comparison using the following phrasing:

“Current models based on ecological theory fail to capture complex community behaviors due to higher-order interactions, do not scale well with increasing complexity and in considering multiple functions.”

and

“We show that the LSTM model can outperform the widely used generalized Lotka-Volterra model based on ecological theory.”

147: A gLV model is linear in the parameters so it is trivial to solve for the coefficients of the model in seconds. The text should be changed here. Also why does it take 5 hours to learn the gLV model when you state in [56] that you have this down to minutes?

We thank the reviewer for asking for additional clarification regarding this point. Reiterating our response above:

“While [11] describes a method for estimating gLV parameters that takes only minutes, this method used only individual species and pairwise community growth data. In our work described in [6], we found that including communities with higher numbers of species reduced the sparsity of the matrix computations and complicated the use of this method. Therefore, we adapted to use the MATLAB FMINCON function. This approach is fully described in [6] and does require hours for parameter estimation.”

Figure 1 (a) is this the model or the teacher forcing training procedure depicted. Also there is a subscript i, suggesting that there is no microbe-microbe interaction information captured by the model. Is the data in figure 1 all forecasted from t=0 or is this 1 sample look ahead prediction performance, it was not clear from the text?

We apologize for the confusion with the figure schematics. This is a depiction of the model. We have updated the instances of *X_i_* in both Figure 1a and Figure 2a to be X with an underscore to clarify that the model input is a vector of species abundances rather than a single species abundance. Thus, this enables the model to capture inter-species microbial interactions. We have updated Figure 1 and Figure 2 captions to clarify this point. All model predictions in the paper are forecasted from time 0, we have added the following sentence to the figure captions for Figures 1 and 2 to clarify this point:

“All predictions are forecasted from the abundance at time 0.”

Figure 2 (a) also has i subscript, was this intentional.

We thank the reviewer for this comment. Reiterating our response from above:

“We apologize for the confusion with the figure schematics. We have updated the instances of *X_i_* in both Figure 1a and Figure 2a to be X with an underscore to clarify that the model input is a vector of species abundances rather than a single species abundance, thus enabling the model to capture inter-species microbial interactions.”

Methods:– The LSTM model is not described in enough detail (what are the dimensions of the objects, how many hidden states [and why], how did you tune the hyperparameters of the model and the gradient solver [what gradient method did you use and why]).

We apologize for any confusion. The model details for each experiment, including the details on the number of hidden states, learning rate for the optimizer, batch size and other hyperparameters were provided in Supplementary File 1. However, we agree that there was inadequate explanation provided and we have corrected this in the revision. We used Adam optimizer for all our experiments. The choice of Adam over simple gradient descent, rmsprop or any other gradient-based optimizers was purely motivated by its empirically fast convergence behavior and lower training and validation errors. For additional details on hyperparameter tuning, please refer to our response 5 to Reviewer #1. For increased clarity, we have added a new section titled “Hyperparameter tuning for LSTM Networks” in the revised paper.

“Similar to other learning algorithms, training an LSTM network entails choosing a set of hyperparameters for optimal performance. We used an exhaustive grid-search for hyperparameter optimization, while the choice of learning algorithm (optimizer) was restricted to Adam due to its superior empirical performance. For each experiment, nearly 10% of the training dataset was reserved as a cross-validation set, and the performances of the trained models were evaluated using cross-validation sets. This information was used to select the best hyperparameter settings. The choices for hyperparameters include: (a) learning rate, (b) number of hidden layers per LSTM unit, (c) number of units per layer within an LSTM unit, (d) mini-batch size, and (e) input data normalization. The input features are normalized to have zero mean and unit variance. The choices of learning rates include 0.005, 0.001 and 0.0001, respectively, each with a decay of 0.25 after every 25 epochs. The gradual decay in learning rates prevents potential overfitting to the data. Choices for mini-batch size included 1, 10, 20 and 50, respectively. It was observed that sizes 10 and 20 resulted in improved training loss, and hence we used mini-batch sizes of 10 or 20 in all evaluations. The number of hidden layers per LSTM cell was iterated from 1 to 2. A two-layered LSTM did not result in any noticeable improvement over a single-layered LSTM cell, and thus we restricted our focus to just a single-layered LSTM cell for the sake of simplicity and faster training/inference. Finally, we tried 512, 1024, 2048 and 4096 hidden units per LSTM cell, and depending upon the complexity of the problem, we used 2048 hidden units (predictions of species and no prediction of metabolites) or 4096 hidden units (simultaneous prediction of species and metabolites). The exact details on the number of training epochs, learning rates, decay rates for different experiments can be found in the Supplementary File 1.”

And

“Note that both the composite as well as the LSTM model require tuning of hyperparameters for optimal performance. The details on the computational implementation are provided in the Methods section.”

– How was the composite model trained, was it indeed in a two-step fashion whereby the gLV model was trained and then a separate inference procedure was run for the metabolite model. More detail is needed here to understand what exactly was done and how the models were trained.

The models are trained in a two-step fashion. The gLV-model being a parametric ODE-model, the model parameters are obtained through nonlinear optimization methods. First, we would like to emphasize again, as we have addressed elsewhere in the review, that we did not train a new gLV model for the composite model in this work. Rather we used the parameters determined in our previous publication [6]. We did train a new regression model for each of the 4 metabolites, because the previous work only trained a model to predict one metabolite (butyrate). We have updated the Methods sections associated with the composite model to clarify this point. Here is that methods text for your reference:

“Our composite model implementation is similar to the model described in [6] for predicting metabolite concentration from community composition at a particular time. We used the exact gLV model parameter distributions from [6] which were determined using an approach based on [11]. The regression model mapping endpoint species abundance to metabolite concentrations from [6] was focused specifically on the prediction of butyrate. Therefore, we adapted the approach to prediction of multiple metabolites. First, we modified the model form to include first order and interaction terms for all 25 species, rather than just the butyrate producers. Then, we separately trained 4 regression models, one for each metabolite (butyrate, lactate, acetate, succinate), using the measured species abundance and measured metabolite concentrations from the same dataset used to train the LSTM model. We trained these models as described previously [6] by using Python scikit-learn [12] and performed L1 regularization to minimize the number of nonzero parameters. Regularization coefficients were chosen by using 10-fold cross validation. We selected the regularization coefficient value with the lowest median mean-squared error across the training splits.”

Plots in general:– Plots only really show how the most abundant taxa are performing, would be nice to see the performance on the lower abundance taxa (maybe take a log of the OD?).

While we appreciate the interest in low abundance taxa, we are not confident in the precision of the experimental measurements of low abundance species (i.e. (*<* 1%) of the total) due to limited sequencing depth. We typically sequence on the order of 10,000 molecules of PCR amplified DNA of the 16S rRNA gene per sample. Therefore, if a species is only represented by on the order of 10 of those molecules (0.1%), then a single sequencing read error would be a 10% error, whereas a species represented by *>* 100 reads (1%) would only have 1% error per read. We chose not to include plots with log scale as to not mislead the reader in regards to the precision of our measurements. We have modified the methods section text to acknowledge this limit on the precision of our measurements:

“We expect the precision of our measurements to drop rapidly for species representing *<* 1% of the community due to limited sequencing depth. We typically sequenced on the order of 10,000 molecules of PCR amplified DNA of the 16S rRNA gene per sample, so if a species is only represented by on the order of 10 of those molecules (0.1%), then a single sequencing read error would be a 10% error, whereas a species represented by (*>* 100) reads (1%) would only have 1% error per read.”

References:You seem to be missing some important references from the Gerber Lab, "MDSINE Microbial Dynamical Systems INference Engine for microbiome time-series analyses" https://doi.org/10.1186/s13059-016-0980-6 is one such example but there are likely more given that lab primarily focuses on microbiome dynamics.

Thanks for the suggestion. We have now cited the Gerber Lab paper in our Introduction.

“The parameters of the gLV model can be efficiently inferred based on properly collected absolute abundance measurements and can provide insight into significant microbial interactions shaping community assembly.”

References

1. Goodfellow, I., Bengio, Y., Courville, A. and Bengio, Y. *Deep learning*, vol. 1 (MIT press Cambridge, 2016).

2. Lundberg, S. M. and Lee, S.-I. A unified approach to interpreting model predictions. In Proceedings of the 31st international conference on neural information processing systems, 4768–4777 (2017).

3. Jia, R. et al. Towards efficient data valuation based on the shapley value. In The 22nd International Conference on Artificial Intelligence and Statistics, 1167–1176 (PMLR, 2019).

4. Jia, R. et al. An empirical and comparative analysis of data valuation with scalable algorithms. arXiv preprint arXiv:1911.07128 (2019).

5. Venturelli, O. S. et al. Deciphering microbial interactions in synthetic human gut microbiome communities. Molecular systems biology 14, e8157 (2018).

6. Clark, R. L. et al. Design of synthetic human gut microbiome assembly and butyrate production. Nature communications 12 (2021).

7. Hromada, S. et al. Negative interactions determine clostridioides difficile growth in synthetic human gut communities. Molecular systems biology 17, e10355 (2021).

8. Venturelli, O. S. et al. Deciphering microbial interactions in synthetic human gut microbiome communities. Molecular Systems Biology 14 (2018). URL https://onlinelibrary.wiley.com/doi/10.15252/msb.20178157.

9. Wang, S. et al. Massive computational acceleration by using neural networks to emulate mechanism-based biological models. Nature communications 10, 1–9 (2019).

10. Marino, S., Baxter, N. T., Huffnagle, G. B., Petrosino, J. F. and Schloss, P. D. Mathematical modeling of primary succession of murine intestinal microbiota. Proceedings of the National Academy of Sciences 111, 439–444 (2014).

11. Shin, S., Venturelli, O. S. and Zavala, V. M. Scalable nonlinear programming framework for parameter estimation in dynamic biological system models. PLOS Computational Biology 15, e1006828 (2019). URL https://dx.plos.org/10.1371/journal.pcbi.1006828.

12. Pedregosa, F. et al. Scikit-learn: Machine learning in python. the Journal of machine Learning research 12, 2825–2830 (2011).

[Editors’ note: further revisions were suggested prior to acceptance, as described below.]

Reviewer #3 still finds that there is a need for a direct "apples-to-apples" comparison of gLV and LSTM. In your response to the reviewer's initial request, you are arguing that such a direct comparison would result in a very low performance of gLV. If this is the case, it would simply strengthen your argument and support the use of LSTM even more. It also did not seem like much additional work to carry out this analysis. If this is done and added as a supplement with some explanation of why it's not in the main text, I believe this would be satisfactory to all reviewers. Other than that, the reviewers were very appreciative of the modifications and the current quality of the paper.

Thank you very much for your efforts in soliciting reviews on the revised draft. We appreciate your acknowledgement of our efforts towards addressing the comments from the reviewers in the last submitted version. We have further revised our manuscript to address the remaining comment from Reviewer#3. Contrary to the opinion of Reviewer 3, an “apples-to-apples” comparison of the gLV and LSTM approaches is not straightforward to implement, and thus in our previous response we took the position that we had adequately demonstrated the benefits of LSTM in our simulations and experiments. However, we have carefully reconsidered our position in light of your comments, and those of Reviewer 3, and we have now included an apples-to-apples comparison between the LSTM and gLV models. While performing this comparison involved quite a bit of work, it demonstrates that the LSTM has better predictive performance for species abundance and metabolites, which indeed strengthens the paper. In the interest of transparency and reproducibility, we have uploaded the full python script of our implementation onto our project Gitlab page.

We believe that our accompanying responses, along with the additional set of experiments suitably addresses the comments from Reviewer#3.

Reviewer #1 (Recommendations for the authors):Thank you. This is exciting work, and the authors have addressed all of my concerns thoroughly.

Thank you very much for your valuable inputs and encouraging remarks.

Reviewer #2 (Recommendations for the authors):The authors have fully addressed my comments on the original submission. I support its publication in the journal.

Thank you very much for your valuable inputs and recommendation for publication.

Reviewer #3 (Recommendations for the authors):One outstanding issue with the paper remains. There is no apples-to-apples comparison of the LSTM+NN to a gLV+NN. I now comment on the authors' responses to an original comment.[Authors] "We did not estimate new gLV parameters for this work, but rather used the gLV parameters that were already determined in [6]."Having a composite model trained on different data and comparing it to a jointly trained model on entirely different data is not a sufficient comparison.

We understand your concern. We have now trained an discrete gLV (approximate gLV) + FFN model for predicting temporal changes in species abundance and metabolite concentrations. In addition, we uploaded the script on the project’s Gitlab page for transparency and reproducibility.

[Authors] "While we agree that the reviewer's proposed method for determining the gLV parameters would be a more direct comparison, there are several reasons why we chose to use the gLV parameters estimated in our previous study: (1) Higher resolution time-series measurements are needed to determine the exponential growth rate in the gLV model. Thus, estimating gLV parameters without the time-series measurements of monocultures would result in a poorly predictive model due to high uncertainty in the exponential growth rate parameters of the gLV model. Therefore, the proposed method of inferring parameters of the gLV model would yield a poorly predictive model."If this is the case then the authors should show that the gLV model is not predictive with this sparse data. This begs the question though. Why did one need the LSTM model if the data doesn't really capture complex dynamics? The abstract stated that the LSTM was necessary for this very reason [Abstract]"Current models based on ecological theory fail to capture complex community behaviors due to higher-order interactions, do not scale well with increasing complexity and in considering multiple functions. We develop and apply a long short-term memory (LSTM) framework to advance our understanding of community assembly and health-relevant metabolite production using a synthetic human gut community." The authors are simultaneously arguing that the dynamics are too complex for simple gLV dynamics while also stating that the new time series are too simple to reliably train a gLV model. Is this a contradiction?

The gLV model captures interactions up to second-order. On the other hand, LSTMs do not impose such restrictions on the type of interactions. Since nonlinear recurrent neural network models, such as LSTMs, have universal approximation guarantees, LSTM networks are capable of learning interactions beyond second-order. This is not a contradiction, but a restatement of the fact that such simple ecological models are only good approximations when the model is close to the underlying ground truth model. This is highlighted in our analysis on simulated data, where mild third-order perturbations significantly reduce the gLV model’s predictive capabilities (Figure 1c).

"(2) The methods used to estimate the gLV model parameters in our previous work were informed by higher resolution time-series measurements of monocultures (individual species). Thus, the gLV model had additional information beyond that of the LSTM model, which was only informed by the initial point and endpoint (with the exception of Figure 5)."This statement is confusing. The LSTM model was trained using teacher forcing with intermediate time points per the methods section?

The gLV model comprises of two components – (a) unary interactions (i.e. exponential growth rate), (b) pairwise interactions. These are described by the following set of coupled-ODEs:

dxi(t)dt=(ri+∑j=1Nspeciesaij xj(t))xi\ (t),

where {*r_i_*} and {*a_ij_*} represent individual species exponential growth rate and pairwise interaction coefficients, respectively. A better suited choice of training procedure for gLV comprises of learning the unary coefficients {*r_i_*} and self-interaction coefficients {*a_ii_*} first, followed by learning the pairwise interaction coefficients {*a_ij_*}*,i* / = *j*. In the absence of any other species, the effect of the term *x_j_*(*t*) · *x_i_*(*t*) is zero, and hence the monocultures (individual species) are informative of the unary coefficients {*r_i_*} and self-interaction coefficients {*a_ii_*}. Using this model framework, a higher time-resolved data will provide a better estimate of the model parameters. We adopted the same procedure for learning {*r_i_*} and {*a_ii_*}. On the other hand, the LSTMs are not built on any explicit first and second-order interactions, and thus, training the LSTM does not require higher time-resolved monospecies growth measurements.

[Re: teacher forcing]: We apologize for any confusion concerning our use of the word ”higher resolution,” which refers to the training phase of LSTM and not to its inference phase. When training an LSTM model, while a component of the loss function is used to force the predictions of the intermediate LSTM units to predict the correct species abundance (and/or metabolite profile) at subsequent time instants, a small disagreement in the species abundance (and/or metabolite profile) at an earlier time point is bound to have a trickle down effect on species abundance (and/or metabolite profile) at later time points. Thus, instead of solely relying on the prediction of the previous LSTM unit, teacher forcing frequently replaces the output of the previous LSTM unit with the ground truth species abundance (and/or metabolite profile) as an input to the next unit. This allows the model to effectively do away with the effects of error in prediction by previous LSTM units, and learn to better advance the prediction forward in time. Recall that teacher forcing is only used during training. During the inference or the test phase, only the initial point abundance (and/or metabolite profile) is rolled forward in time using the learned model assisted by teacher forcing.

"(3) The optimization algorithm MATLAB FMINCON is a widely used approach for estimating parameters of the gLV model (see [6], [8] and [10], for example) (4) The proposed method of log transforming the gLV model only works with higher resolution time-series data. We have sparse time sampling in our datasets."If indeed the data is so sparse that a traditional gLV model would not learn then it would be easy for the authors to show this. In order to be published Figure 4 should compare gLV+NN and LSTM+NN trained on the same data in the same way and compare the model's performance directly.

We appreciate the reviewer’s suggestion to include the apples-to-apples comparison. We implemented a variation of the reviewer’s proposed discretized gLV model, which is described in the Methods section under Comparison of the Discretized gLV Model to the LSTM. We provide two comparisons between gLV+NN and LSTM+NN, where both models were trained on the same data using the same training method to predict species abundance and metabolite concentrations. In the first comparison, we apply both models to predict species abundance after training on in silico data that was generated from a ground truth gLV model. The in silico species abundance data was generated from a ground truth gLV model with randomly sampled growth rates and interaction coefficients according to r_j_ ∼ N(µ = .36,σ = .16), a_i,i_ ∼ N(µ = −1.5,σ = .25), and a_i,j_ ∼ N(µ = −.22,σ = .33). A densely sampled dataset was sampled at one hour intervals over a period of 48 hours, and a sparsely sampled dataset was sampled at 16 hour intervals. For each dataset, a total of 200 runs were simulated, each with a unique initial condition generated using Latin hypercube sampling with values ranging from.001 to.1. 75% of the data was used for training, with the remaining 25% held-out for testing. While the approximate gLV model performs well to predict species abundances after training on densely sampled data, it performs poorly compared to the LSTM model when trained on sparsely sampled data. We present the results from this comparison as a Jupyter notebook, which is available on the manuscript’s GitLab repository.

In the second comparison, we compared a discretized gLV model that was augmented with a feed-forward neural network to enable both species and metabolite predictions so that the model could be directly compared to LSTM model M3. Because the results shown in Figure 4 are based on data that was sampled at two time points, we compared the discretized gLV model to the LSTM for the time-series dataset showin in Figure 5, which was trained on data with observations at four time points to provide a better comparison of each model’s ability to predict system dynamics. In Figure 5—figure supplement 3, we show that the LSTM outperforms the gLV in both species abundance prediction and metabolite concentration prediction. Our results from both comparisons reinforced our main conclusions that the LSTM is capable of achieving improved performance over the discretized gLV model both trained using the same algorithm.